

# Carbonyl sulfide (OCS) exchange between soils and the atmosphere affected by soil moisture and compensation points

Rüdiger Bunk[1, #], Zhigang Yi[1,2, #], Thomas Behrendt[3], Dianming Wu[1,4], Meinrat Otto Andreae[1], Jürgen Kesselmeier[1]

[1]Max Planck Institute for Chemistry, Biogeochemistry and Multiphase Departments, Mainz, Germany
[2]Fujian Agriculture and Forestry University, College of Resources and Environment, Fuzhou, China
[3]Max Planck Institute for Biogeochemistry, Biogeochemical Processes Department, Jena, Germany
[4]Key Laboratory of Geographic Information Sciences, Ministry of Education, School of Geographic Sciences, East China Normal University, 200241 Shanghai, China
# These authors contributed equally to this work

Correspondence to: Jürgen Kesselmeier (j.kesselmeier@mpic.de)

**Abstract.** Carbonyl sulfide (OCS) is a chemically quite stable gas in the troposphere (lifetime ~2-6 years) and consequently some of it is transported up to the stratosphere where it contributes to the stratospheric sulfate layer. Due to the similarities in uptake mechanism between OCS and $CO_2$, the use of OCS as a proxy for $CO_2$ in ecosystem gross primary production (GPP) has been proposed. For this application a good understanding of uptake ($U_{OCS}$) and production ($P_{OCS}$) processes of OCS in an ecosystem is required. A new OCS quantum cascade laser coupled with an automated soil chamber system enabled us to measure the soil-atmosphere OCS exchange of four different soil samples with high precision. The adjustment of the chamber air to different OCS mixing ratios (50, 500, and 1000 ppt) allowed us to separate production and consumption processes and to estimate compensation points (CPs) for the OCS exchange. At an atmospheric mixing ratio of 1000 ppt, the maximum $U_{OCS}$ was of the order of 22 to 110 pmol $g^{-1}$ $h^{-1}$ for needle forest soil samples and of the order of 3 to 5 pmol $g^{-1}$ $h^{-1}$ for an agricultural mineral soil, both measured at moderate soil moisture. Uptake processes ($U_{OCS}$) were dominant at all soil moistures for the forest soils, while $P_{OCS}$ exceeded $U_{OCS}$ at higher soil moistures for the agricultural soil, resulting in net emission. Hence, our results indicate that in (spruce) forests $U_{OCS}$ might be the dominant process, while in agricultural soils $P_{OCS}$ at higher soil moisture and $U_{OCS}$ under moderate soil moisture seem to dominate the OCS exchange. The OCS compensation points (CPs) were highly dependent on soil water content and extended over a wide range of 130 ppt to 1600 ppt for the forest soils and 450 ppt to 5500 ppt for the agricultural soil. The strong dependency between soil water content and the compensation point value must be taken into account for all further analyses. The lowest CPs were found at about 20% water filled pore space ($WFPS_{lab}$), implying the maximum of $U_{OCS}$ under these soil moisture conditions and excluding OCS emission under such conditions. We discuss our results in view of other studies about compensation points and the potential contribution of microbial groups.

*Key words:* carbonyl sulfide (OCS); soil; water content; OCS flux; OCS compensation point



## 1. Introduction

Carbonyl sulfide (OCS) is the most abundant sulfur-containing trace gas in the atmosphere. Montzka et al. (2007) report a globally averaged tropospheric mixing ratio of 476±4 ppt for the Northern Hemisphere, which is in good agreement with previous studies. OCS is an important contributor to stratospheric sulfate aerosol (SSA; Crutzen 1976; Barkley et al., 2008), which affects the radiative balance of the atmosphere (Turco et al., 1980; Charlson et al., 1987) and plays a role in the ozone-depleting chemistry (Fahey et al., 1993; Roche et al., 1994; Solomon et al., 1996). In view of the potential role of OCS, the investigation of its sources and sinks has been intensified and recent findings demonstrate that the stratospheric cooling effect by SSA is balanced by the greenhouse effect in the troposphere (Brühl et al., 2012).

The role of terrestrial ecosystems as the largest sink for tropospheric OCS has been studied for more than 20 years (Chin and Davis, 1993; Kettle et al., 2002; Montzka et al., 2007), but uncertainties of the estimate of the sink strength were still quite large. Vegetation was considered as the main sink for OCS, and the close relationship between OCS uptake and photosynthetic $CO_2$ uptake lead to a considerable increase of the sink strength estimates based on GPP (Sandoval-Soto et al., 2005). Furthermore, changes of the sink strength of vegetation as a response to global change is a matter of discussion (White et al., 2010; Sandoval-Soto et al., 2012).

The relationship between concentrations of OCS and gross primary production (GPP) has been explored as a tool for constraining estimates of fluxes in both the carbon and sulfur cycles (Asaf et al., 2013; Berry et al., 2013; Billesbach et al., 2014; Blonquist et al., 2011; Campbell et al., 2008; Montzka et al., 2007; Sandoval-Soto et al., 2005; Seibt et al., 2010; Suntharalingam et al., 2008). The uptake of OCS from the atmosphere is thought to be dominated by the activity of carbonic anhydrase (CA), an enzyme abundant in leaves, which also catalyzes $CO_2$ hydration during photosynthesis (Protoschill-Krebs and Kesselmeier, 1992; Protoschill-Krebs et al., 1996; Notni et al., 2007). The OCS taken up by leaves undergoes hydration catalyzed by CA, which leads to the virtually irreversible formation of hydrogen sulfide ($H_2S$) and is regarded as a unidirectional exchange. In contrast, the $CO_2$ flux is bidirectional (i.e., the net uptake measured is the result of gross uptake and respiratory release). With this background, the ratio of $CO_2$ to OCS uptake by vegetation is discussed as a useful tracer of GPP, the largest flux in the global biogeochemical carbon cycle (Sandoval-Soto 2005; Seibt et al., 2010; Asaf et al., 2013; Berry et al., 2013; Berkelhammer et al. 2014, Campbell et al., 2008, 2017). The complex background for this discussion has recently been summarized in a synthesis paper (Whelan et al. 2017).

However, the use of OCS as a proxy for GPP is based on the assumption that there are only minor other sources and sinks in the ecosystem besides the vegetation. If there are other large sources or sinks of atmospheric OCS in an ecosystem, this approach will become as complex as for $CO_2$. Soils were originally treated as a source of OCS because early observations were often made in chambers with an initially sulfur-free headspace, which led to artificially high OCS emissions and the estimation of around 21–25% of the total global source (Khalil and Rasmussen, 1984; Chin and Davis, 1993; Johnson et al., 1993). In contrast, more recent field or laboratory work using enclosures employing ambient air as a sweep gas found that





soil acted as a sink (Castro and Galloway, 1991; De Mello and Hines; 1994; Kesselmeier et al., 1999; Kuhn et al., 1999; Simmons et al., 1999; Steinbacher et al., 2004; Yi et al., 2007; Van Diest and Kesselmeier, 2008; Bunk et al., 2017).

Although the understanding of soils as a major sink helps to explain the "missing sinks" for OCS, soil uptake still shows a wide scatter among different environments. Also, the drivers of soil OCS fluxes and their dependences to environmental parameters (such as soil moisture, soil temperature, and OCS mixing ratio) are still largely unknown. Uncovering the mechanisms for soil OCS fluxes would allow soil-atmospheric OCS exchange to be estimated on broader spatial scales. This ability would benefit both atmospheric sulfur cycle studies and efforts to estimate ecosystem GPP using OCS as a proxy of $CO_2$ uptake.

Many environmental parameters were found to influence the soil-atmosphere OCS exchange, especially temperature and soil water content (Goldan et al., 1987; Kesselmeier et al., 1999; Van Diest and Kesselmeier, 2008). Some researchers found an optimum temperature for soil OCS fluxes between 16 and 20°C (Lehmann and Conrad, 1996; Kesselmeier et al., 1999; Van Diest and Kesselmeier, 2008). Furthermore, there is a strong evidence that the OCS exchange between soil and the atmosphere is dependent on the ambient OCS mixing ratio (Lehmann and Conrad 1996, Kesselmeier et al. 1999). Conrad (1994) discussed the theoretical background of a compensation mixing ratio or compensation point (the mixing ratio where the trace gas production and uptake are balanced and the net exchange equals zero) for various trace gases. Based on this background and their soil-atmosphere OCS exchange measurements at different ambient OCS mixing ratios, Lehmann and Conrad (1996) have calculated compensation mixing ratios for four different soils.

Soil water content also plays an important role in soil-atmosphere OCS exchange, and the optimum soil water content for OCS uptake depends on the type of soil (Kesselmeier et al., 1999; Van Diest and Kesselmeier, 2008; Whelan et al., 2016). Agricultural soils have been characterized as either an OCS source or sink (Bunk, et al., 2017; Whelan et al, 2015; Maseyk et al., 2014), whereas forest soils have been characterized as sinks (Sun et al., 2017; Steinbacher et al., 2004). Furthermore, Bunk et al. (2017) reported recently that the $CO_2$ mixing ratio, which may be high in soil pores, has a considerable effect on soil-atmosphere OCS exchange and may shift the exchange to a pronounced emission.

To deepen our understanding of source and sink characteristics, we analyzed the OCS exchange rates, production rates ($P_{OCS}$), uptake rates ($U_{OCS}$), consumption rate coefficient ($k_{OCS}$), and compensation points (CP) (see section 2.3) in relation to soil water content and OCS atmospheric mixing ratios for two organic layers (Oh) and one litter layer (L) from needle forests and compare our laboratory approach for the litter layer sampled in 2012 to a chamber based field study in 2015 (Sun et al., 2017). Furthermore, we analyzed one soil sample from an agricultural field (currently wheat), which had been already investigated earlier (Kesselmeier et al., 1999; van Diest and Kesselmeier, 2008) and which we consider to be representative for a mid-latitude agricultural ecosystem.



## 2 Material and methods

### 2.1 Soil samples

Soil samples were collected from four sites: (1) An agricultural soil from a field near Mainz, Germany, which had been previously planted with wheat (49.95 ºN, 8.25 ºE), a site that had been also studied by Kesselmeier et al. (1999) and Van Diest and Kesselmeier (2008). Sites (2) and (3) are located within a spruce forest, Waldstein, Germany. Both consist of organic layers (Oh), with (2) originating from a forest whose understory was dominated by blueberries (50.1420 ºN, 11.8665 ºE) and (3) by young spruce trees (50.1425 ºN, 11.8673 ºE). The fourth sample (4) was from a Scots pine litter layer at the Hyytiälä site, Finland (61.846 ºN, 24.295 ºE). Samples were taken from the top 5 cm from multiple grabs (up to 5) at all sites in order to account for the variability within a single location on the site. We tried to use the same method to collect all soil samples, but we cannot exclude the variability over time. Fresh subsamples of agricultural soil in Mainz were oven dried (at 40° C) for comparison. In the following, samples of the Mainz soil that were stored in an oven-dried state will be referred to as "Mainz dry", samples that were stored at field moisture as "Mainz fresh". All samples were sieved with a stainless steel sieve with a mesh size of 2 mm (Mainz soil) or 16 mm (organic soils) and were stored in polyethylene bags at 5°C until analyzed. Total carbon, total sulfur, total nitrogen, $NO_3$-N, $NH_4$-N content, and pH were determined by an external company (Enviliytix, Wiesbaden, Germany) and are summarized in Table 1. The maximum water holding capacity (MWHC) was determined by moistening with deionized water (R 18.2 MΩ) according to conventional methods.

### 2.2 Experimental setup

All measurements were performed with the automated dynamic chamber system set up by Behrendt et al. (2014). Briefly, it comprises of a set of Plexiglas soil chambers in a temperature-controlled dark incubator that can be flushed with a controlled amount of a gas mixture. The flushing gas was controlled by a set of valves and mass flow controllers (Bronckhorst, Germany). Samples were drawn at the outlets of the individual chambers. Additional mass flow controllers were introduced to mix OCS and $CO_2$ at desired mixing ratios into the flushing air.

The soil chambers were filled with either 80 g of Mainz soil (mineral, Ap horizon) or 20 g of the organic horizons of three forest soils, and then the soils were wetted to nearly 100 % $WFPS_{lab}$ (Water Filled Pore Space as determined described in 2.4) with deionized water (R 18.2 MΩ). One chamber remained empty as a control. The chambers were flushed with about 2.5 l min$^{-1}$ of dry compressed air that had been passed through a pure air generator (PAG 003, Ecophysics, Switzerland) beforehand. OCS and $CO_2$ mixing ratios in the inlet flushing gas were adjusted to about 400 ppm for $CO_2$ and 50, 500, or 1000 ppt for OCS by addition of those gases in respective amounts by mass flow controller from standard gas cylinders (10 % $CO_2$, Westfalen, Germany) and (500 ppb OCS, Air Liquide, Germany). Samples were drawn by the analyzer units from the outlet of individual chambers to determine the trace gas mixing ratios (including OCS [LGR OCS/CO Analyzer, Los



Gatos Research, USA], $CO_2$, $H_2O$ [Licor 860, LiCOR, USA]). Sampling was switched between the sample chambers by computer controlled valves (see Behrendt et al., 2014). Performance and calibration of the new OCS analyzer are described in detail in Bunk et al. (2017).

5 **2.3 OCS release rates, $P_{OCS}$, $U_{OCS}$, deposition velocity and compensation points**

Exchange rates, $E_{OCS}$ [pmol $g^{-1}$ $h^{-1}$], were calculated based on the difference in OCS concentration between the sample and reference chamber, $\Delta_{OCS}$ [pmol $mol^{-1}$], the flushing rate, Q [mol $h^{-1}$], and the amount of soil in the sample chamber, $m_{soil}$ [g], according to Equation (1).

$$E_{OCS} = \Delta_{OCS} \times \frac{Q}{m_{soil}} \tag{1}$$

10 Additionally, exchange rates, $E_{OCS,A}$ [pmol $m^{-2}$ $s^{-1}$], were calculated according to Equation (2).

$$E_{OCS,A} = \Delta_{OCS} \times \frac{Q}{A_c} \tag{2}$$

where $\Delta_{OCS}$ [pmol $mol^{-1}$] is the difference in OCS concentration between the sample and reference chamber, Q [mol $s^{-1}$] is the flushing rate, and A [$m^2$] is the sample chamber area.

Soil moisture changes were derived from the amount of water vapor released by the soil sample (integration of the difference 15 between sample and reference chamber water vapor concentration over time) according to Behrendt et al. (2014).

The OCS compensation point (CP) is the OCS mixing ratio at a given soil moisture at which OCS uptake ($U_{OCS}$) and production ($P_{OCS}$) are balanced and the net exchange (exchange rate, $E_{OCS}$) is zero. CP can be calculated by the following process: The exchange rate, $E_{OCS}$ at a given soil moisture can be expressed as

$$E_{OCS} = P_{OCS} + U_{OCS} \tag{3}$$

20 where $P_{OCS}$ is the OCS production and $U_{OCS}$ is the OCS uptake. The $P_{OCS}$ for any ambient OCS mixing ratio will be equal to the net exchange rate at an ambient OCS mixing ratio of zero ppt at the corresponding soil moisture. This is based on the linear relationship between OCS uptake and OCS mixing ratio shown by Kesselmeier et al. (1999) and the assumption that the ambient OCS mixing ratio does not influence OCS production (see 4.1). The $P_{OCS}$ was calculated according to equation (4). Finally, a CP for a given soil moisture can then be calculated according to Equation (3), where the exchange rate $E_{OCS}$ 25 equals zero pmol $g^{-1}$ $h^{-1}$. The consumption rate coefficient, $k_{OCS}$, is the slope of the regression between the exchange rate at 50 ppt and 1000 ppt ambient OCS mixing ratio and c is the OCS mixing ratio.

$$E_{OCS} = P_{OCS} + k_{OCS} \times c \tag{4}$$

The deposition velocity, $V_d$, was calculated based on the OCS exchange rate, $E_{OCS,A}$ [pmol $m^{-2}$ $s^{-1}$], and the ambient OCS concentration, c [pmol $m^{-3}$], according to Equation 5.




$$V_d = \frac{E_{OCS,A}}{c}$$
(5)

**2.4 Water Filled Pore Space WFPS$_{lab}$**

For the determination of the soil moisture, the mass of soil water was converted into water filled pore space, WFPS$_{lab}$ [%], according to Bourtsoukidis et al. (submitted) by

$$WFPS_{,lab}(t_i) = \frac{m_{soil}(t_i) - m_{soil}(t_s)}{m_{soil}(t_s)} \cdot \frac{100}{\theta_s}$$
(6)

where $\theta_s$ is the saturated gravimetric water content in the laboratory at the beginning of the experiment and $m_{soil}(t_s)$ equals the mass of soil at the end of the experiment, respectively. The value of $\theta_s$ was determined experimentally for each homogenized soil sample (sieved through a 2mm mesh) followed by the addition of H$_2$O until the surface of particles was covered by a thin film of water. The soil moisture of the Finland soil was recalculated into units of m$^3$ m$^{-3}$ using the density

of water at 20 ºC (0.998 g cm$^{-3}$) and the bulk density of this soil (0.1 g cm$^{-3}$) reported in Pumpanen and Ilvesniemi 2005.

**2.5 Accuracy, precision, and limit of distinguishable OCS exchange**

As the calculation of the exchange rate E is based on the difference of two measured mixing ratios (mixing ratio in the sample and reference chambers, respectively), the noise of both measured mixing ratios adds up. Therefore, it is reasonable

to set a threshold below which such a difference cannot reliably resolve the exchange rates. Parallel to classical limit of detection calculations we defined this resolution threshold, $t_R$, as three times the noise of the analyzer. This threshold corresponds to an exchange rate of ±1 pmol g$^{-1}$ h$^{-1}$ for the Mainz soil and ±3 pmol g$^{-1}$ h$^{-1}$ for the forest soils. Exchange rates that are smaller than these values cannot be safely discerned from a zero exchange.

Instrument accuracy and limitations are discussed in detail in Bunk et al. (2017). In summary, precision was found to be

better than 5 ppt when measuring mixing ratios from a source with a static OCS mixing ratio. Accordingly, this corresponds to a flux rate precision of 0.8 pmol g$^{-1}$ h$^{-1}$ (Mainz soil) or 3.3 pmol g$^{-1}$ h$^{-1}$ (forest soils). Accuracy was determined with permeation sources or certified gas mixtures and showed excellent matching over a wide range of mixing ratios. Only the NOAA-standard with a typical atmospheric mixing ratio (449.2 ppt ± 1.4 ppt, Essex stainless steel cylinder, cylinder number. SX-3584, NOAA, USA) was underestimated by 7%. Therefore, calculated fluxes may be underestimated by 7%

with no significant impact on the conclusions of this work.



## 3 Results

### 3.1 OCS exchange for organic and litter layers from needle forests in comparison to an agricultural soil

The exchange of OCS between the four soils and the atmosphere at 50 and 1000 ppt ambient OCS is shown in Figure 1. All four soils showed OCS uptake at medium soil moisture when ambient OCS was high (1000 ppt). The uptake was reduced or

switched to emission at high and low soil moistures. The two Waldstein soils had a rather broad uptake peak with a maximum at around 40 % $WFPS_{lab}$. OCS uptake at medium soil moisture was 20 % stronger for the Waldstein soil with its young spruce understory than for the one with a blueberry understory.

In comparison to the other soils, the uptake maxima of the Finland and Mainz soils were sharper and located around 18 % and 20 % $WFPS_{lab}$, respectively. At higher soil humidity, the Mainz soil emitted OCS, while the emission was very low for

the Finland soil. The maximal net uptake ($E_{OCS}$ minimum) varied between soils, ranging from 3 pmol $g^{-1}$ $h^{-1}$ (Mainz soil) to 13 pmol $g^{-1}$ $h^{-1}$ (Waldstein Blueberry) and 23 pmol $g^{-1}$ $h^{-1}$ (Waldstein Spruce) and to 85 pmol $g^{-1}$ $h^{-1}$ (Finland Needle Forest) as shown in Figure 1. The corresponding calculated total uptake rates ($U_{OCS} = E_{OCS}-P_{OCS}$, see 2.3) are 5, 25, 32 and 110 pmol $g^{-1}$ $h^{-1}$ respectively.

At low ambient OCS mixing ratio (50 ppt), all soils showed OCS emission that was mostly constant at any soil moisture,

except for some decline at very low soil humidity. Emission strength varied between soils, ranging from about 1 pmol $g^{-1}$ $h^{-1}$ (Waldstein Blueberry) over 2 pmol $g^{-1}$ $h^{-1}$ (Mainz Soil) and 3 pmol $g^{-1}$ $h^{-1}$ (Waldstein Spruce) up to 15 pmol $g^{-1}$ $h^{-1}$ (Finland Needle Forest).

### 3.2 OCS exchange of fresh and dry Mainz soil

The OCS exchange correlated strongly with the ambient OCS mixing ratio and soil humidity. The net exchange is shown in Figure 2. For the fresh Mainz soil at high OCS mixing ratio (1000 ppt), the exchange behavior followed the basic pattern of emission-uptake-emission (from wet to dry soil), as already observed by Bunk et al. (2017). At 500 ppt ambient OCS mixing ratio, the uptake in the medium humidity range was reduced in comparison to the uptake at 1000 ppm OCS. Emission at high and low humidity was similar to the 1000 ppt experiment. While OCS emission (at 500 ppt ambient mixing ratio) was about

2 pmol $g^{-1}$ $h^{-1}$ at high soil moisture, uptake at medium soil moisture and production at low soil moisture were below the resolution threshold (1 pmol $g^{-1}$ $h^{-1}$, see Section 2.5) for the dry/fresh Mainz soil and therefore cannot be accurately distinguished from a zero exchange. At 50 ppt ambient OCS, there was a nearly constant emission of OCS of about 2 pmol $g^{-1}$ $h^{-1}$. For the Mainz soil that had been air dried before storage, the exchange patterns were similar, but a general decrease of the OCS emission rates was found. OCS releases at low and high humidity, as well as under low ambient OCS mixing ratios,

were lower. Uptake in the medium humidity range was stronger, especially at high ambient OCS.





### 3.3 OCS compensation points

The OCS compensation points were found to be variable in close dependence on the soil water content. The Mainz soil (dry and wet storage), Finland litter layer, and Waldstein spruce soil showed high compensation points for wet and dry soil and lower compensation points at a range of humidity between the wet and dry extremes (see Figure 3). The CPs were in the
range of 300 to 5500 ppt, 130 to 320 ppt, 180 to 1150 ppt and 210 to 1730 ppt for Mainz soil, Waldstein blueberry soil, Waldstein spruce soil, and litter layer Finland soil, respectively. The Mainz soil and litter layer Finland soil had their lowest CP in the moisture range of roughly 15 and 40 % $WFPS_{lab}$. The Waldstein blueberry soil had its highest CP in the extremely dry range and in the moisture range of 73 to 80% $WFPS_{lab}$, while the other 3 soils had their highest CP in the extremely dry and extremely wet moisture ranges.

### 3.4 $P_{OCS}$, $U_{OCS}$ and $k_{OCS}$ in relation to soil moisture

The behavior of $U_{OCS}$, $P_{OCS}$ and the corresponding exchange rates as a function of soil moisture are shown in Figure 4. $P_{OCS}$ did not vary significantly with soil moisture, except for the Waldstein soil with blueberry understory. For this soil, $P_{OCS}$ increased slightly at moderate soil moisture. The values of $P_{OCS}$ were 2, 4, 7 and 30 pmol $g^{-1}$ $h^{-1}$ for Mainz soil, Waldstein
soil with blueberry or young spruce understory, and Finland needle forest litter, respectively. On the other hand, $U_{OCS}$ was strongly influenced by soil moisture, with a maximum at medium soil moistures and lower $U_{OCS}$ at low and high soil moisture. The maximum/minimum $U_{OCS}$ were 5/0, 25/5, 33/8 and 120/20 pmol $g^{-1}h^{-1}$ for Mainz, Waldstein Blueberry, Waldstein Spruce soil and Finland litter layer, respectively. It is important to note, that the consumption rate coefficient ($k_{OCS}$) is not constant, but changes with the decrease in soil moisture (see Figure 5). The change in $k_{OCS}$ with soil moisture is
rather similar that of $U_{OCS}$. This might indicate involvement of multiple OCS uptake processes (see section 4.2).

## 4. Discussion

### 4.1 OCS exchange for organic layers and litter layer from needle forests

The OCS exchange from our laboratory measurements of the Finland litter layer soil is of the same magnitude as the field
OCS exchange measurements performed by Sun et al. (2017) at the site where our samples were taken (SMEAR II site, Hyytiälä), despite different measurement methods, experimental conditions and the fact that the samples for our laboratory study had been collected two years earlier. Only the data measured at temperatures between 15 °C and 16.1 °C were selected from Sun et al. (2017). To account for the temperature gap of approximately 5° C between our data and that of Sun et al. (2017), the deposition velocities of our lab measurements were corrected based on the temperature optimum curve presented
in Kesselmeier et al. (1999) by a factor of 0.852 (the ratio of OCS uptake at 15° C to OCS uptake at 20° C in Kesselmeier et





al., 1999). While our exchange measurements showed a clear dependence of OCS exchange on soil moisture, the exchange rates from the field measurements of Sun et al. (2017) are more scattered over the soil moistures they measured at, and the relationship between the exchange rate and the soil moisture is less clear. However, all the exchange rates measured by Sun et al. (2017) fall within the exchange rates observed in our lab measurements. We suggest that the stronger scatter in the field

measurements is due to additional factors that were kept constant in our lab measurements, but unavoidably vary during field measurements. Figure 6 shows a comparison of the OCS deposition velocities (uptake rate normalized by ambient OCS mixing ratio, see 2.3). This suggests that, (1) laboratory measurements with soil chambers as performed in this study can adequately simulate processes at field sites, and (2) the OCS exchange at the SMEAR II site is dominated by processes in the litter layer.

All three organic forest soil samples were almost exclusively OCS sinks, with $U_{OCS}$ being much higher than $P_{OCS}$, especially at moderate soil moisture, a behavior which may change under elevated $CO_2$ concentration, however (see Bunk et al., 2017). The $U_{OCS}$ and net uptake fluxes were a lot higher than for the agricultural mineral soil examined in this study. The litter layer sample from the Finland site (SMEAR II Station, Hyytiälä) had both significantly higher $P_{OCS}$ and maximal $U_{OCS}$ than the

two Waldstein samples from the organic layer (roughly 4-fold on average each). Like the good agreement of our laboratory exchange measurements with the field data from Sun et al. (2017) (see above) this suggests that the litter layer might be the most important layer for soil-atmosphere OCS exchange. Furthermore, in experiments utilizing the selective inhibitor Nystatin we observed that fungi might play a dominant role in OCS uptake (Bunk et al., 2017). Important differences in the vertical distribution of a soils' fungal community have been reported. Lindahl et al. (2007) describe a vertical distribution of

fungi in needle forest soil, with saprotrophic fungi preferring the upper litter layer and mycorrhiza fungi the deeper litter layer and organic horizon. Dickie et al. (2002) report spatial variation in the abundance and distribution of mycorrhiza fungi with different groups of fungi preferring different depths in the litter layer and soil. This stratification in microbial community may lead to important differences in OCS exchange behavior of different soil layers, as discussed below.

In general, carbonic anhydrase (CA), which is abundant in most heterotrophic and autotrophic organisms, is assumed to

consume OCS (Notni et al., 2007, Blezinger et al., 1999, Protoschill-Krebs et al., 1996). We specifically selected a representative mid-latitude mineral soil and, in contrast, organic-rich soil horizons from a spruce forest to investigate the effect of autotrophic and heterotrophic life-forms on OCS consumption. Heterotrophs (saprotrophic fungi and mycorrhiza) are commonly more dominant at elevated total carbon (see Table 1) in the organic rich forest soils compared to agricultural soils, which are limited in organic carbon. The higher abundance and activity of heterotrophs in the organic forest soils

might explain the wider range of $U_{OCS}$ with respect to soil moisture. The difference in carbon content is likely due to the different vegetation in the understory, changing the input of organic C into the soil.

Autotrophic bacteria (Ogawa et al., 2013, Kato et al., 2007, Seefeld et al., 1995) and archaea (Smeulders et al., 2011), which are capable of fixing $CO_2$ from the atmosphere, are also known to simultaneously consume OCS. These organisms are





commonly more abundant in organic limited soils, such as agricultural soils (However, see Section 4.4 for the case of the Finland litter sample and available carbon). Because CA is considered ubiquitous, some OCS consumption is expected for any microorganism, if the ambient mixing ratio is high enough. Therefore, even though there is evidence suggesting that fungi might dominate OCS consumption, especially in forest soils rich in organic C, other groups of microorganisms can also be expected to contribute to the observed consumption. Further disentanglement of this mixture of uptake signals would require additional tools like a molecular approach, which would have exceeded the feasible scope of this work. However, these methods have been employed in a follow-up study (Behrendt et al., submitted.), indicating involvement of autotrophs such as ammonium oxidizing bacteria (AOB) and some methanotrophs in OCS exchange. As heterotrophs and autotrophs are likely to be involved in soil OCS exchange (also see section 4.4), it can be expected that more than one process is involved in each soil's OCS uptake signal. Indeed, we found at least two distinctive uptake processes by linking OCS and CO fluxes in a follow-up study (see Behrendt et al., submitted, for detail). At higher and lower soil moisture their activity is reduced. The remaining $U_{OCS}$ is either due to a different uptake process or the remaining but reduced activity of the main consumers.

The value of $P_{OCS}$, on the other hand, does not vary strongly with soil moisture. This indicates that either the organisms producing OCS are generalists in respect to soil moisture or that the bulk of OCS production in the soils examined here is not of biotic nature. Both $U_{OCS}$ and $P_{OCS}$ are higher for the forest soils than for the agricultural Mainz soil, and $P_{OCS}$ increases with the carbon content of the soil, although the correlation is not linear. Both $P_{OCS}$ and $U_{OCS}$ can benefit from high organic matter contents in the soil. One reason is that OCS can be produced from thiocyanate by microorganisms (Katayama et al., 1992, Katayama et al., 1998), whose main source in soils is decomposition of plant material. Also, heterotrophic consumers and producers of OCS alike will benefit from organic carbon as an energy source in the soil. Additionally, organic material in the ground should expected to contain at least some sulfur containing compounds, which would be required for all known and unknown OCS production pathways.

As the vast majority of data points for OCS exchange of the forest soils do exceed the resolution threshold described in section 2.5, this threshold is generally not of relevance for the discussion of forest soil OCS exchange. One exception is the OCS exchange of the Waldstein spruce soil, where most data points are below the resolution threshold of 3 pmol h$^{-1}$ g$^{-1}$, which would put the calculation of $P_{OCS}$ and consequently the CPs for this soil in question. However, some points are above the threshold for all soil moistures and the exchange follows a solid trend that is also consistent with the exchange of the other 3 soils (at 50 ppt ambient OCS mixing ratio) examined in this study. Therefore, we consider the observed values as reliable.





## 4.2 OCS exchange of fresh and dry Mainz soil

Wet soils have shown to tend towards emission of OCS (e.g., Minami and Fukushi 1981, Davai and Delaune 1995, Yang et al. 1996) while well aerated soils tend to take up OCS (Castro and Galloway 1991, Kuhn et al. 1999, Watts et al. 2010). Abiotic emission has been reported for very dry soils (Whelan et al. 2016, Whelan and Rew 2015). This is in agreement with

the pattern of OCS exchange from the mid-latitude agricultural soil, which has been observed already in Van Diest and Kesselmeier (2008), Bunk et al. (2017) and within this study. As $P_{OCS}$ does not change significantly with soil moisture, the changes in $E_{OCS}$ must be driven by $U_{OCS}$. This suggests that the enzymatic processes responsible for OCS uptake are mainly active at moderate soil moisture (20 % $WFPS_{lab}$) for this soil. We suspect these processes to be connected to autotrophic organisms, as discussed in Section 4.4. The organisms related to the production of OCS on the other hand must be active at

nearly the full range of tested soil moisture (except very low soil moisture). As expected, $U_{OCS}$ was strongest when the ambient OCS mixing ratio was highest (1000 ppt) and weakest when the ambient mixing ratio was lowest (50 ppt). The exchange rate at 500 ppt ambient mixing ratio was intermediate between the 50 and the 1000 ppt exchange. This is in clear accordance with the assumption of a linear dependence of $U_{OCS}$ on the ambient OCS mixing ratio, and thus validates the approach by which CPs were calculated as described in section 2.3, and is in good agreement with the findings of

Kesselmeier et al. (1999).

The negative net exchange value at moderate soil moisture and 500 ppt ambient mixing ratio is smaller than the resolution threshold (see Section 2.5), and would therefore have to be considered indistinguishable from a zero exchange. However, the OCS exchange relative to soil moisture, both for fresh and previously dried Mainz soil, very clearly follows the same trend of emission-uptake-emission at 500 and 1000 ppt, with the expected decrease of $U_{OCS}$ at the lower mixing ratios. Therefore,

the switch between net emission and net uptake for Mainz soil at 500 ppt is most likely a true trend (as the exchange values for the 1000 ppt experiment exceed the resolution threshold) despite the exchange for the 500 ppt experiment being below the resolution threshold. Aside from that, only the experimental data obtained at 50 ppt and 1000 ppt were used to calculate the compensation points.

## 4.3 OCS compensation points (CPs)

The CP of a soil at a given soil moisture is determined by the corresponding $P_{OCS}$ and $U_{OCS}$. As $P_{OCS}$ does not change much with soil moisture (see Figure 4), the observed change of the CP with soil moisture is mainly driven by $U_{OCS}$.

According to our calculations (see Figure 3) we expect the Waldstein soils and the Finland needle leaf soil to act as sinks. This is in agreement with the Sun et al. (2017) field measurements identifying the soil as a sink. With CP between 460 to

510 ppt at about 20-30% $WFPS_{lab}$, which is near the typical atmospheric OCS mixing ratio, the agricultural Mainz soil is expected to be a weak sink at this moderate soil moisture. At moistures above and below that moisture range (low soil moisture and high soil moisture range, see Bunk et al., 2017, Figure 4) Mainz soil is expected to be a source as its CP are





higher than the typical atmospheric OCS mixing ratio in these moisture ranges. Seasonal fluctuations as well as nearby strong sinks or sources can modify local OCS mixing ratios, shifting a soil's expected OCS exchange behavior accordingly.

Some compensation points have been calculated in previous works (Lehmann and Conrad 1996; Kesselmeier et al., 1999) and are compared to our results in Figure 3. Lehmann and Conrad (1996) found very high compensation points when including measurements at very high ambient OCS mixing ratios, while their calculations yielded lower, much more realistic compensation points when only including data from experiments at lower ambient mixing ratios. They proposed that this discrepancy was due to another uptake mechanism that was only active at very high OCS mixing ratios, which might probably be related to another life-form utilizing OCS as substrate (e.g., sulfur oxidizers) not related to the enzymes of autotrophic and heterotrophic $CO_2$ fixation. Three (of four) of the compensation points calculated by Lehmann and Conrad (1996) fall within the range of compensation points we calculated for our soil measurements and are shown in Figure 3. Only the compensation point for a fourth soil was far outside the range of compensation points we observed for soils in our study (approx. 11500 ppt at 18% $WFPS_{lab}$, which is roughly tenfold the highest values we observed at this soil moisture) and is not shown in Figure 3. Though limited, this agreement of compensation points from Lehman and Conrad (1996) with ours would support the existence of two uptake mechanisms (one at typical atmospheric mixing ratios, one at very high mixing ratios) as suggested by Lehmann and Conrad (1996).

Kesselmeier et al. (1999) determined the compensation point for one soil at one soil moisture for several temperatures. This soil is from the same site as the Mainz soil in this work. They found a temperature dependent range of compensation points (57 to 300 ppt atmospheric mixing ratio) that is in the same magnitude as the compensation point we found for this soil at the same soil moisture. However, the compensation points in the upper range in Kesselmeier et al. (1999) are about 100 ppt lower than the ones we observed. The lowest CP Kesselmeier et al. (1999) observed are about 300 ppt lower than the ones we found for the same soil. This range is shown in Figure 3 as a black bar. This difference might be due to changes in the soil that occurred during the 15 years that passed between the two samplings of the site, or due to differences in experimental method and instrumentation.

### 4.4 Relationship of consumption rate coefficient, $k_{OCS}$, and soil moisture

The consumption rate coefficient (k-coefficient) is derived from the slope between a soils OCS exchange at high and low atmospheric mixing ratios for a given soil moisture (see Figure 5). It can be considered to roughly describe the OCS uptake efficiency of the microbial soil community at a given soil moisture. Two distinct sequences of k can be identified when comparing k-coefficients of the soils examined (see Figure 5). The Mainz soil and the Finland soil show highest k-coefficients at approximately 20% $WFPS_{lab}$, sharply decreasing with higher and lower soil moisture. In contrast, the two





Waldstein soils have the highest k-coefficients above 40%, and the decrease in k-coefficients at drier und wetter soil is far less pronounced. We hypothesize that this may indicate two different uptake processes, one with optimal soil moisture of 20% $WFPS_{lab}$ and one with optimal soil moisture of > 40% $WFPS_{lab}$. Based on the selection of organic-rich forest soil (> 40 % C, containing high amounts of readily available carbon) and mineral soil low in organic carbon (2.5 % C, see Table 1),

these might be related to autotrophic and heterotrophic lifestyles. However, the Finland soil, while having the highest carbon content, is a litter layer sample. Consequently, carbon in lignified needles (up to 25% lignin content, Berg et al., 1984) is not easily accessible for many organisms (Albers et al., 2004; Peláez et al., 1995; Schlegel, 1992; Taylor et al., 1989; Kirk 1983), but specialized organisms can break down needles, especially some saprotrophic fungi (Koukol et al., 2006; Peláez et al., 1995; Schlegel, 1992). Furthermore, control over litter decomposition is small in material with small amounts of lignin

but strong when lignin content is high (Taylor et al., 1989). The two Waldstein samples with their broad maximum of the k-coefficient are from the O-layer were the carbon has been weathered and is more available and easy to break down for a wide range of heterotrophs. Therefore, the 20% $WFPS_{lab}$ k-peak may be related to autotrophs and heterotrophs specialized in breaking down lignified needles (note the difference in absolute k between Mainz and Finland soil) that are favored when little or difficult accessible carbon is available, while the > 40% $WFPS_{lab}$ k-peak may be related to heterotrophs utilizing the

more readily available carbon in the O-layers of the Waldstein samples.

One more point that should be considered is that the Finland soil, while having its k-coefficient peak at 20% $WFPS_{lab}$ like the Mainz soil, exhibits k-coefficients at > 40 similar to the two Waldstein samples. This is consistent with the Finland litter layer sample containing carbon that is difficult to break down, favoring both autotrophs and heterotroph specialized in metabolizing lignified structures.

### 4.5 Comparison of the exchange behavior of Mainz soil after dry and moist storage and from a decade earlier

The OCS exchange behavior of the Mainz soil followed a very similar pattern after dry storage and storage with moisture as found while sampling ("fresh"). Samples had been stored 5 to 9 months. For both, there was uptake of OCS at about 12% gravimetric soil water content, which was reduced when the soil contained a higher or lower amount of water, gradually

switching to emission of OCS at wet and very dry states as demonstrated in Figure 7. Exchange measurements made 6 years earlier by van Diest and Kesselmeier (2008) show a very similar curve, but here uptake is stronger and is only reduced in the wet and very dry soil moisture range without switching to emission. This suggests two things:

(1) Even when resampling after 6 years and after two different types of sample processing, the same mechanisms and drivers control the OCS exchange of the Mainz soil, producing very similar trends and patterns. This demonstrates the power of

utilizing the soil moisture response functions for upscaling approaches/modelling purposes.



(2) Some changes occur both over time and are induced by two different ways of sample treatment (storing "fresh" or air drying the sample before storage), illustrating the importance of consistent treatment and storage of samples that are meant to be compared to each other.

### 4.6 Comparison of plant-atmosphere OCS exchange to soil-atmosphere OCS exchange at the example of a spruce forest

Our OCS exchange rates for spruce forest soil as well as exchange rates for another spruce forest soil reported by Steinbacher et al. (2004) are low (about 1-5%) compared to the average fluxes over a spruce forest reported by Xu et al. (2002). Uptake rates of spruce forest soils and spruce are summarized in Table 2. Considering field data, soil OCS fluxes for spruce forest soils appear to be small and negligible when conditions for OCS uptake by plants are favorable. However, forest OCS uptake can be highly variable (Xu et al., 2002), because plant OCS uptake is controlled by stomatal conductance (Sandoval Soto et al., 2005). Stomatal aperture in turn is influenced by factors as temperature, light and water availability. Consequently, for spruce forests, when stomatal conductance is low due to lack of light, drought or other unfavorable circumstances, soil OCS exchange might represent a significant portion of ecosystem OCS exchange due to plant OCS exchange being reduced. Additionally, the ratio of soil-atmosphere OCS exchange to plant-atmosphere OCS exchange will strongly depend on the leaf area index (LAI) of an ecosystem. LAI describes the ratio of leaf surface area per ground surface area (Asner et al., 2003). For example, the mean LAI for temperate evergreen needle leaf forests is reported to be 5.5, with maximum values of 15. Sandoval-Soto et al. (2005) found plant-atmosphere OCS uptake of 12.6 pmol $m^{-2}$ $s^{-1}$ for spruce trees in chamber measurements (note that $m^{-2}$ for the Sandoval-Soto et al. (2005) data refers to square meter leaf surface as opposed per square meter ground surface as in the rest of this manuscript). Applying the LAI compiled by Asner et al. (2003) to the exchange rate of spruce trees measured by Sandoval-Soto et al. (2005) would yield expected OCS exchange rates of 69.3 pmol $m^{-2}$ $s^{-1}$ (average canopy density), 0.12 pmol $m^{-2}$ $s^{-1}$ (very low canopy density) or 189 pmol $m^{-2}$ $s^{-1}$ (high canopy density). The ratio of spruce forest soil uptake compared to plant uptake at average LAI would then be roughly 1% or 5% when comparing soil-atmosphere exchange from Steinbacher et al. (2004) or from this work to the projected mean plant-atmosphere exchange. Considering the span between the maximum and minimum LAI values reported in Asner et al., however, the influence of soil-atmosphere exchange ranges from minimal to dominant. This suggests that soil uptake might be negligible in spruce forests when conditions for OCS uptake by plants are favorable and LAI is average or higher. However, if LAI is low or when conditions are otherwise not favorable for OCS uptake, soil exchange can introduce a substantial error if not considered. As similarly wide ranges of LAI are reported for other biomes (Asner et al., 2003) this might hold true for other biomes as well.



## 5 Conclusions

The high $P_{OCS}$ and $U_{OCS}$ of the Finland litter layer in comparison to $P_{OCS}$ and $U_{OCS}$ from the organic layer of the Waldstein soil show that soil-atmosphere OCS exchange is mainly driven by the litter layer at the Hyytiälä site. This is supported by the

good agreement of $E_{OCS}$ measured for litter layer Finland soil with field data from Sun et al. (2017) who measured at the same sampling site. Furthermore, these similarities support that results from laboratory chamber measurements as performed in this study can adequately simulate field conditions and results can be transferred to the origin site.

The composition of the fungal community in forest soils appears to play a major role in soil OCS uptake. Saprotrophic fungi

in the litter layer and mycorrhiza fungi in the organic rich horizons might be involved in the OCS uptake.

The relationship between soil moisture and $E_{OCS}$ is mainly driven by $U_{OCS}$, suggesting a high dynamic in $U_{OCS}$ by potentially different enzymatic processes within soils as they are drying out.

Comparison of exchange rates for spruce forests and spruce soils suggests that soil-atmosphere OCS exchange might be negligible in spruce forests when conditions for plant OCS uptake are favorable, but might constitute a significant fraction of ecosystem OCS exchange when conditions are not favorable for plant OCS uptake.

The observed switch of the agricultural Mainz soil from OCS uptake at medium soil moisture to emission at high and low

soil moisture appears to be typical for agricultural soil and is in good agreement with prior publications. This result may indicate a variable behavior of non-forest soils with respect of their role as sources or sinks of OCS, which may have an impact on global OCS budget estimation. The differences in k-rates suggest this variability might be related to autotrophic and heterotrophic life-forms.

The exchange of OCS with soils includes a compensation point, as reported earlier (Lehmann and Conrad, 1996; Kesselmeier et al., 1999). The compensation points are highly dependent on soil moisture and thus variable. This is complicating the analysis of exchange processes. Differences of the compensation point values can be understood as a result of soil water content.

Comparison of exchange rates for Mainz agricultural soil that was either stored field fresh or after being oven dried while otherwise being subjected to the same experimental procedures (including rewetting) show that different storage methods may lead to variations in the behavior of measured soil samples. It is important to be aware of this effect and to choose the storage method to be used in advance and in accordance with the objectives of the study.



**6 Acknowledgements and data**

Data is available as data file (Bunk_Yi_comppoints_DATA.zip) in the supplement. This research was funded by the German Max Planck Society and the National Natural Science Foundation of China (Grant NO. 41473083, 41173090). Dianming Wu

was supported by "the Fundamental Research Funds for the Central Universities".

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





**Table 1. Properties of the tested soils**

| Sample ID | Ecosystem | NH$_4$-N | NO$_3$-N | TC | TN | TS | pH | MWHC |
|-----------|-----------|----------|----------|-----|-----|-----|-----|------|
| | | [mg/kg] | [mg/kg] | [%] | [%] | [%] | - | [g g$^{-1}$] |
| Mainz fresh (Germany) | Wheat field | <0.04 | 0.85 | 2.5 | 0.17 | 0.03 | 7.6 | 0.7 |
| Mainz dry (Germany) | Wheat field | 1.5 | 0.87 | 2.5 | 0.16 | 0.02 | 7.5 | 0.7 |
| Waldstein blueberry (Germany) | Spruce forest | 240[a] | 37[a] | 40 | 1.9 | 0.26 | 3.2[a] | 3.8 |
| Waldstein spruce (Germany) | Spruce forest | 983[a] | 90[a] | 45 | 2.1 | 0.26 | 3.0[a] | 3.7 |
| Finland scots pine forest (Finland) | Scots pine forest | 1.6[b] | 2.0[b] | 47 | 1.4 | 0.17 | 3.0[b] | 8.2 |

MWHC: maximum water holding capacity.

a: the data is from Behrendt et al. (2014) and b: the data is from Oswald et al. 2015.

**Table 2: OCS uptake by spruce forest soil and spruce trees in our measurements and reported in literature**

| Spruce forest soil uptake | | | |
|---------------------------|--|--|--|
| Source | uptake pmol m$^{-2}$ s$^{-1}$ | Method | Comment |
| This work | 2.98 | Chamber measurement, lab | Blueberry understory, max. uptake at optimal soil moisture |
| This work | 3.42 | Chamber measurement, lab | Young spruce understory, max. uptake at optimal soil moisture |
| Steinbacher et al., 2004 | 0.81 | Chamber measurement, field | Average uptake, daytime |
| Spruce plant uptake | | | |
| Xu et al., 2002 | 93 | Relaxed eddy accumulation , field | average |
| Sandoval Soto et al., 2005 | 12.6 | Chamber measurements, lab | 600 µmol m$^{-2}$ s$^{-1}$ light, Reference area (m$^{-2}$) is leaf surface |





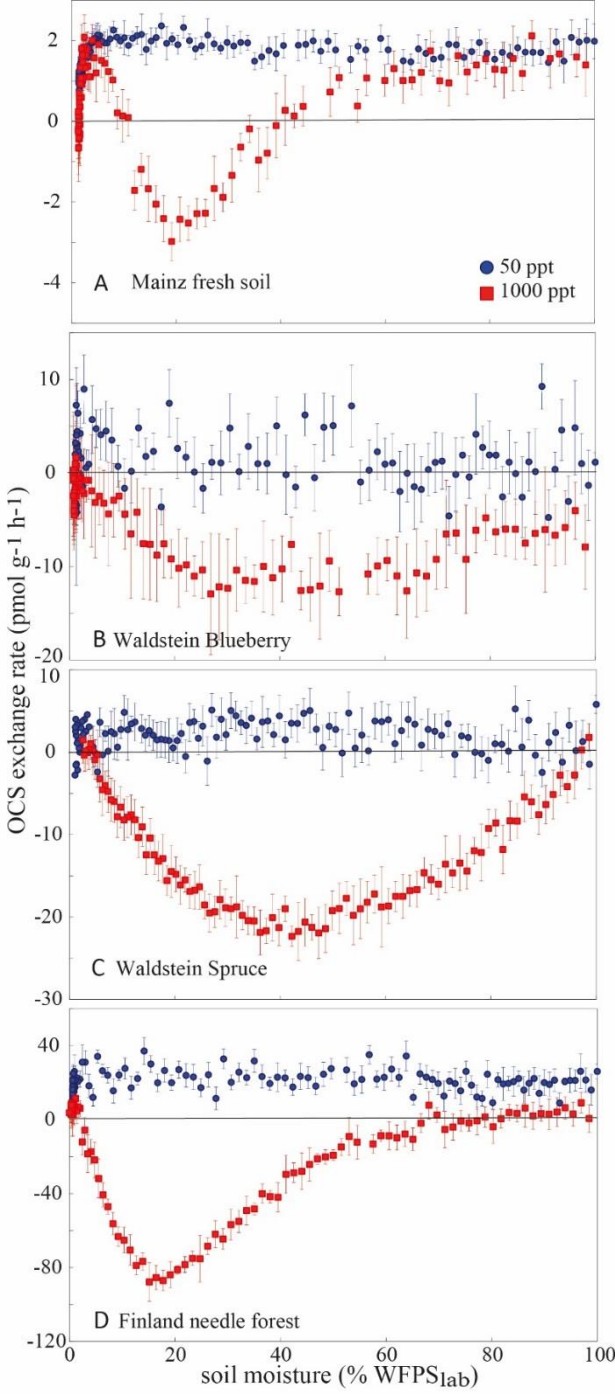

**Figure 1: OCS net exchange rates for different fresh soil samples at 50 and 1000 ppt ambient OCS in relation to soil moisture, WFPS$_{lab}$ (A: Mainz agricultural soil; B: Waldstein blueberry soil; C: Waldstein spruce soil; D: Litter layer Finland needle forest soil).**





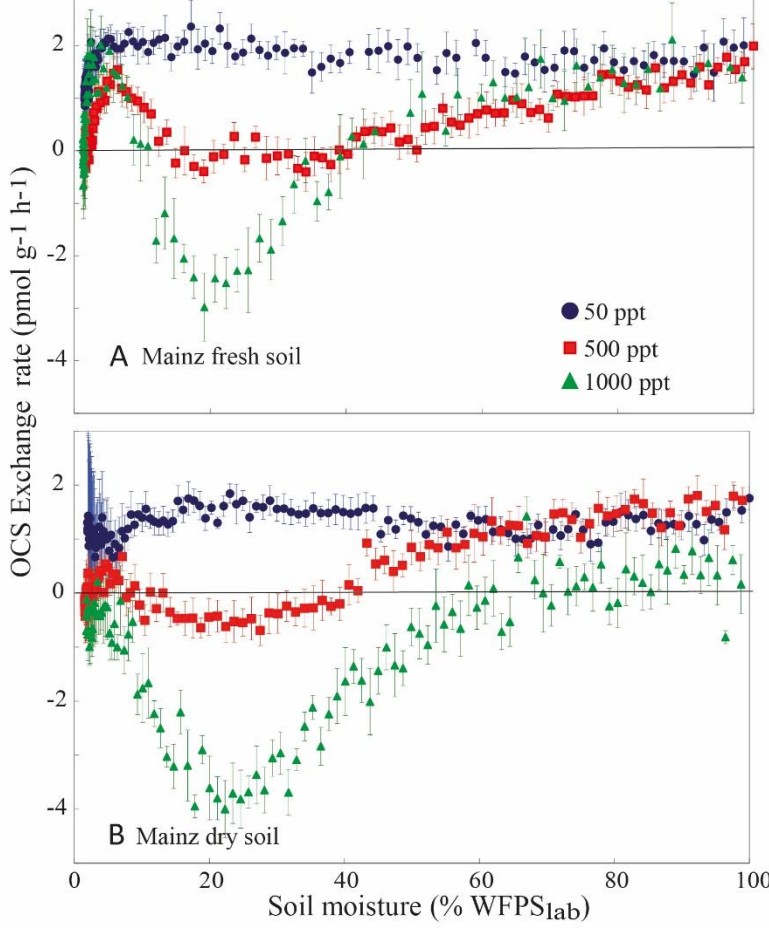

**Figure 2. OCS net exchange rates in relation to WFPS$_{lab}$ at 20º C for 80 g of fresh and dry agricultural soil per cuvette. Different ambient OCS mixing ratios with ~50 ppt (blue cycle), ~500 ppt (red square), or ~1000 ppt (green triangle) OCS in the flushing air**
5  **were applied. (A: fresh soil; B: dried soil)**



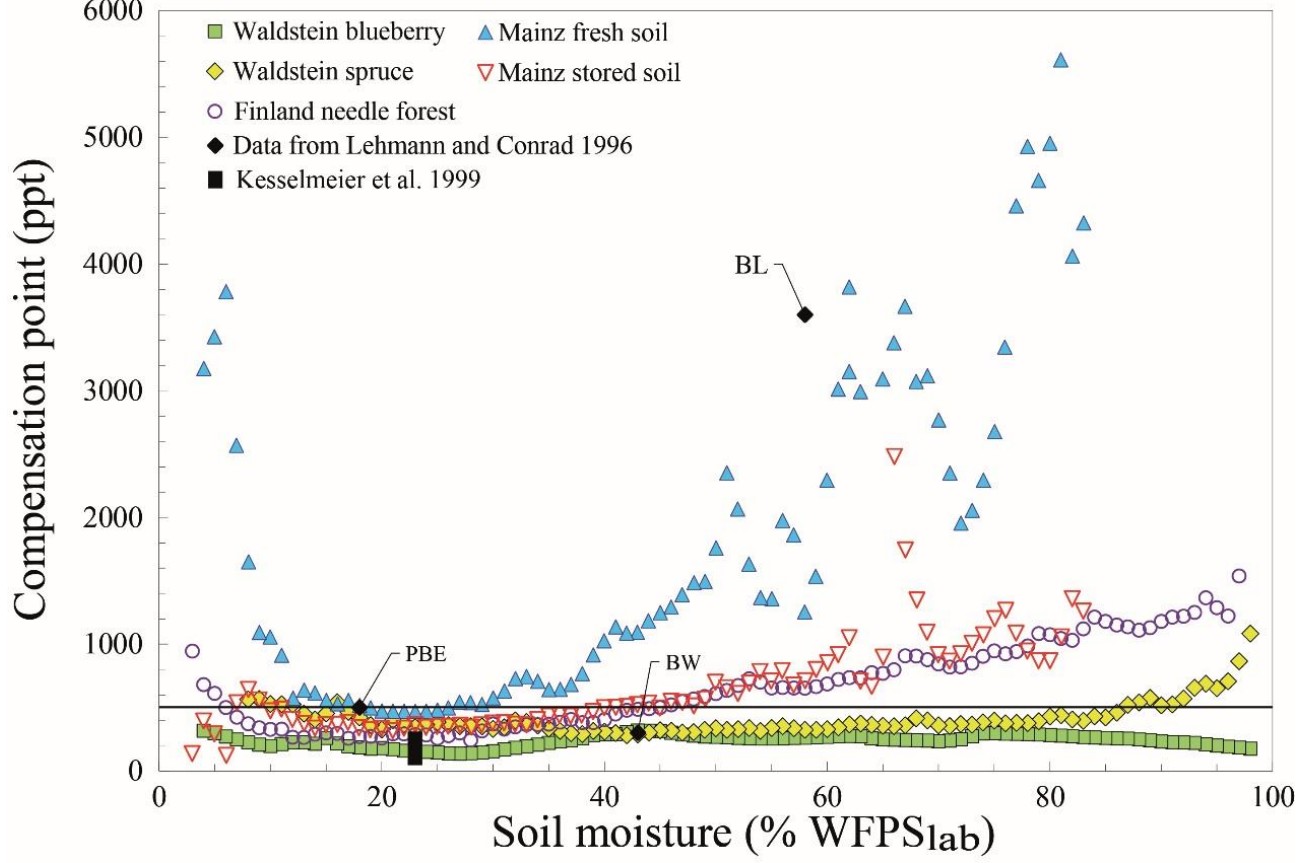

**Figure 3: OCS compensation mixing ratios (CP) of four soils at different soil water contents. CP were calculated based on the net exchange rate under the ambient OCS mixing ratio of ~1000 ppt and ~50 ppt, based on Equations (2) and (3) (see 2.3). For comparison compensation points calculated by Lehmann and Conrad (1996) (see 4.3) were added as black diamonds, compensation points from Kesselmeier et al. (1999) (see 4.3) were added as a black bar. Soils from Lehmann and Conrad are: PBE, a forest soil without small vegetation (Marburg); BW, a forest soil vegetated with *Dryopteris assimilis, Mnium undulatum* and *Leucobryum glaucum* (Schachtenau) and BL, a forest soil vegetated with *Allium ursinum* (Radolfzell). The soils were measured at approximately 43% (PBE), 44% (BW) and 58% (BL) WFPS$_{lab}$. Kesselmeier et al. (1999) determined the compensation point for one soil (taken from the same field as the "Mainz" samples in this study) at different temperatures. The bar shown represents the range of compensation points, with its lower edge representing the lowest and its upper edge representing the highest CP. The continuous black line marks the average tropospheric OCS mixing ratio of the northern hemisphere.**



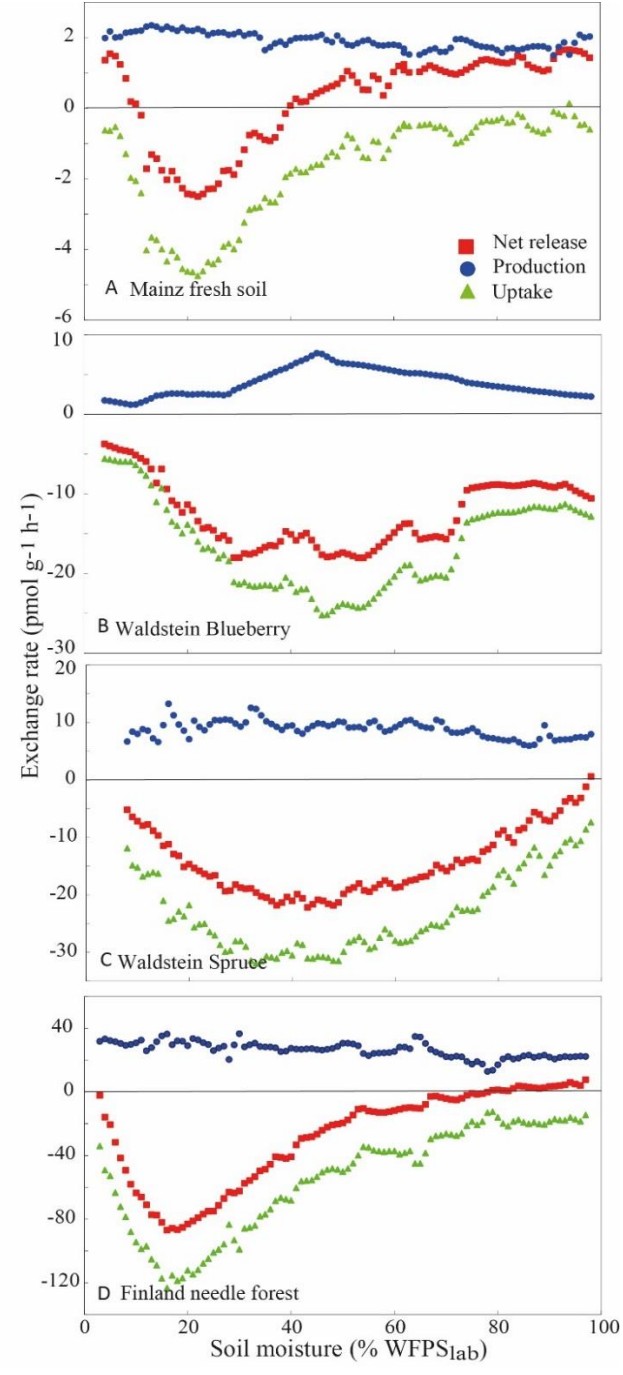

**Figure 4.** $U_{OCS}$ and $P_{OCS}$ calculated as described in section 2.3 and $E_{OCS}$ measured at 1000 ppt OCS mixing ratio for A: Mainz soil, B: Waldstein spruce soil (blueberry understory), C Waldstein spruce forest soil (young spruce understory) and D: Finland needle leaf forest.



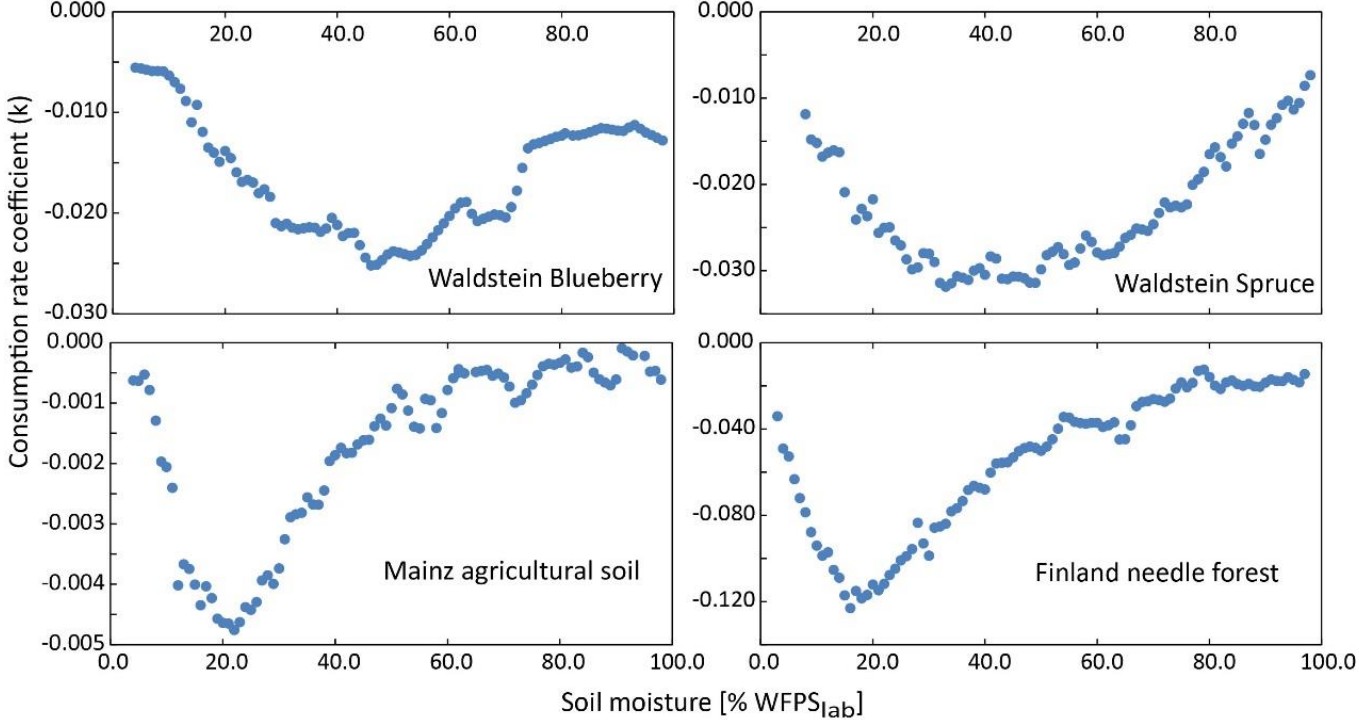

**Figure 5: Consumption rate coefficients (derived from the slope between a soils carbonyl sulfide exchange at low and at high ambient OCS mixing ratio: $E_{OCS} = P_{OCS} + k_{OCS} \times c$; see Section 2.3) calculated as described in Section 2.3 in relation to soil moisture [% WFPS$_{lab}$].**





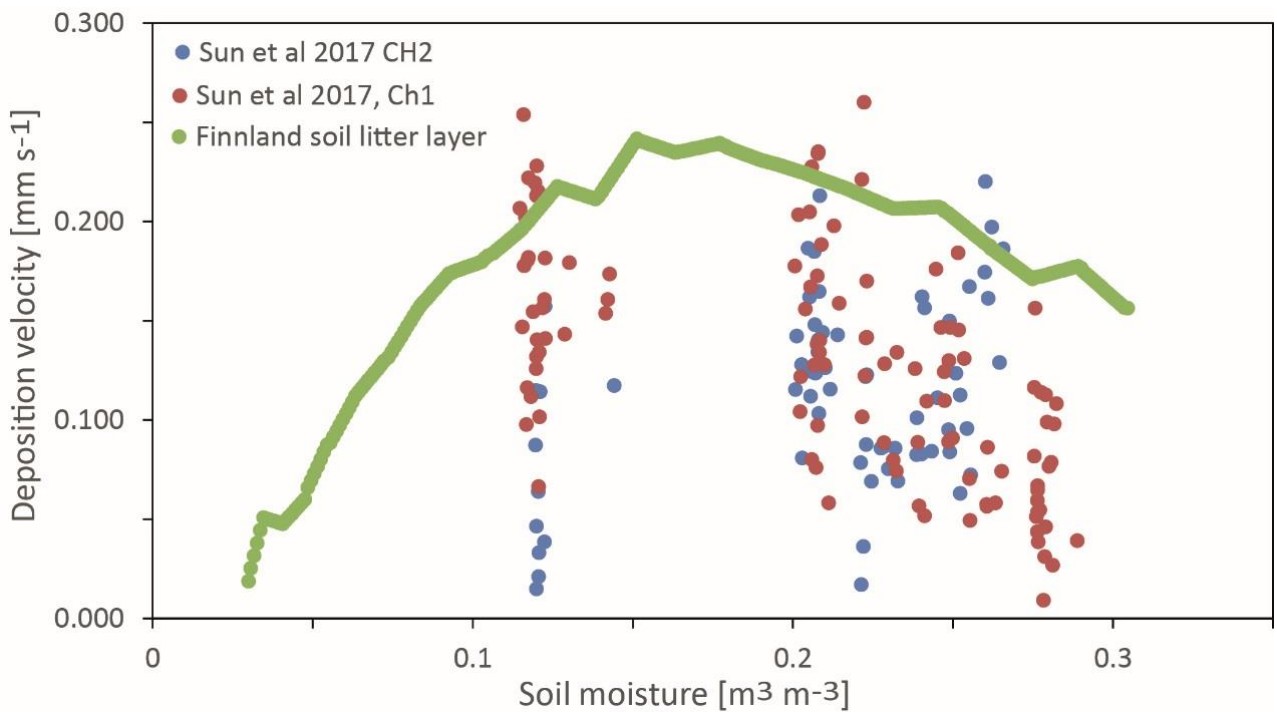

**Figure 6: Field data from Sun et al. 2017 (blue and green dots) compared to our lab exchange measurements. Temperature of the Sun et al. (2017) data is variable, while our measurements used constant temperature. Therefore, only data measured at similar temperature (>15° C and < 20° C) was selected from Sun et al. 2017 for comparison. To compensate for varying OCS mixing ratios during the field measurements we recalculated our data and the data from Sun et al. (2017) into deposition velocities (see 2.3).**





**Figure 7: German agricultural soil from Mainz wheat ecosystem: A: COS uptake rates (pmol g⁻¹ h⁻¹) in relation to the soil water content (% gravimetric soil water content) at 20°C for 80 g of soil per cuvette with the ambient COS mixing ratio of about 1000 ppt as inlet flushing air, compared with the results from Van Diest *et al.* (2008) for 200 g**