# Peer review of "Carbonyl sulfide (OCS) exchange between soils and the atmosphere affected by soil moisture and compensation points"

_Biogeosciences, 2018_

## Referee Comment (RC1) · Anonymous Referee #1 · 15 May 2018

**General comments**

The manuscript presents a laboratory study on two important drivers of soil OCS exchange—soil moisture and ambient OCS concentration. The authors collected four soil samples from the field, incubated them in the laboratory, and determined their OCS exchange patterns under varying conditions of soil moisture and OCS concentrations. The experimental design covered the full range of relative soil moisture, from 0% to 100% with a fine step. This allowed the construction of well-defined soil moisture response curves of OCS exchange. By measuring OCS exchange under different ambient OCS concentrations, this study was also able to separate OCS gross uptake

and production components from the net exchange and investigate their respective responses to soil moisture.

However, the experimental design to derive compensation points was seriously flawed and unable to produce robust estimates of the compensation points. As the authors noted, they used measurements at 1,000 ppt and 50 ppt to derive the compensation points. **This practice is essentially to fit a regression line with only two data points, and then extrapolate to somewhere up to 5,500 ppt!** The uncertainty would likely be huge. Yet they present no uncertainty measures of this derived quantity in the figures or the text. That is not to say that the data shouldn't be published, but the authors should quantify the uncertainties and properly acknowledge the limitations of their method.

The manuscript identified the compensation point as a driver of OCS exchange, as the title and some section titles implied. But a critical examination of the mechanisms governing OCS uptake and production processes suggests that this is a misinterpretation. Compensation point manifests the dynamic balance between uptake and production. It is not an intrinsic property of the soil, but a variable that depends on soil moisture, as the authors have demonstrated in Fig. 3. Therefore, it is inaccurate to say that OCS exchange is 'affected' by the compensation point.

The manuscript suffers from other flaws. The equations (Eqs 1 and 2) used to calculate the OCS exchange apply only to the **steady state** condition, which can be shown mathematically by solving a mass balance equation for chamber gas exchange. Yet, they did not assure the readers of the validity of the steady state assumption used in their measurements. In addition, the interpretations of data are insufficient, and sometimes superficial. I would suggest the authors to exploit their dataset for new understanding rather than confirmatory interpretations. The discussion is poorly organized and lacks novel insights. This part has to be rewritten for clarity and coherence. As a suggestion, please figure out the main point of each section and organize the paragraphs in a streamlined way that helps convey the main point and impart understanding to the

readers. Detailed comments on specific issues are listed below.

**Specific comments**

Abstract

> P1L24–26: "The OCS compensation points (CPs) were highly dependent on soil water content and extended over a wide range of 130 ppt to 1600 ppt for the forest soils and 450 ppt to 5500 ppt for the agricultural soil."

If OCS exchange was measured only at "50, 500, and 1000 ppt" (P1L18), how did you obtain a compensation point of 5,500 ppt for a certain sample? This has not been explained in the manuscript. Presumably, the only way would be to extrapolate a linear relationship between the net exchange and the concentration. The compensation points derived this way will inevitably have large uncertainties, since there were only three (and sometimes just two!) data points to fit a line. Please provide the uncertainty measures and address the limitations of this method.

Introduction

> P2L6: "In view of the potential role of OCS, . . ."

What kind of 'potential role'? Please clarify.

> P2L2–8: "Carbonyl sulfide (OCS) is . . . in the troposphere (Brühl et al., 2012)."

I think the first paragraph of the Introduction can be shortened significantly without losing the sense.

P2L16: "in both the carbon and sulfur cycles"

The 'sulfur cycle' is not a major concern of the studies cited here. I suggest removing it.

P2L15–26: "The relationship between concentrations of OCS and gross primary production (GPP) . . ."

This paragraph is an elaboration on the use of OCS as a tracer for GPP. While broadly speaking this is a motive for many recent studies on soil OCS exchange, as plant uptake of OCS is not the topic of this study, having such a level of details in the Introduction seems excessive. I suggest cutting this paragraph down to three sentences or so.

P3L3–4: "Although the understanding of soils as a major sink helps to explain the 'missing sinks' for OCS, soil uptake still shows a wide scatter among different environments."

Where does the "missing sink" come from, all of a sudden? Please explain.

P3L4–5: "Also, the drivers of soil OCS fluxes and their dependences to environmental parameters (such as soil moisture, soil temperature, and OCS mixing ratio) are still largely unknown."

This is not an accurate statement. Actually, you have cited quite a few studies in the next few paragraphs to show that the drivers are not 'largely unknown.'

P3L5–6: "Uncovering the mechanisms for soil OCS fluxes would allow soil-atmospheric OCS exchange to be estimated on broader spatial scales."

I think that Berry et al. (2013) and Launois et al. (2015) have already estimated soil–atmosphere OCS exchange 'on broader spatial scales'. Please clarify what you mean by this statement.

P3L12–17: "Furthermore, there is a strong evidence that the OCS exchange between soil and the atmosphere is dependent on the ambient OCS mixing ratio . . ."

What's missing from the introduction of the compensation point is the fundamental cause of it: OCS uptake is a pseudo first-order reaction and thus depends linearly on the ambient OCS concentration, whereas OCS production is independent of OCS concentration (Conrad, 1994).

P3L24: "To deepen our understanding of source and sink characteristics . . ."

At the end of the Introduction it is still not clear to the readers what specific research questions this study aims to address. The authors should consider formulating research questions or hypotheses to better orient the readers.

Material and methods

P4L3: "Soil samples were collected from four sites . . ."
P4L9–10: "We tried to use the same method to collect all soil samples, but we cannot exclude the variability over time."

Can you provide information on **when** each sample was collected?

> P4L10–11: "Fresh subsamples of agricultural soil in Mainz were oven dried
> (at 40°C) for comparison."

Please specify how long they had been dried for in the oven.

Typically, using the gravimetric method to determine soil moisture would require samples to be oven dried at 105°C for 48 to 72 hours, but of course that temperature would not be desirable because microbial communities could be destroyed. I assume that the choice of 40°C had something to do with this concern, but at this temperature, how did you ensure that the samples were **dried completely** and the maximum soil moisture ($\theta_s$) used in Eqn (6) was not biased by any remaining water?

> P4L16 and P4L26: "deionized water (R 18.2 M$\Omega$)"

The unit of DI water resistivity is M$\Omega$·cm, not M$\Omega$.

> P5L21–23: "This is based on the linear relationship between OCS uptake
> and OCS mixing ratio shown by Kesselmeier et al. (1999) and the assump-
> tion that the ambient OCS mixing ratio does not influence OCS production
> (see 4.1)."

This sentence explains the theoretical basis of the 'compensation point' and should better be placed in the Introduction.

> P5L9, Eqn (1): $E_{OCS} = Q\Delta_{OCS}/m_{soil}$

This equation works only under a **steady state** condition. How did you ensure that OCS concentration in the chamber headspace reached a steady state? Please provide relevant details and reasoning.

Results

> P7L6–7: "OCS uptake at medium soil moisture was 20% stronger for the Waldstein soil with its young spruce understory than for the one with a blueberry understory."

What range is the "medium soil moisture"? Be quantitative.

Waldstein spruce soil showed a peak uptake of 23 pmol $g^{-1}$ $h^{-1}$ and Waldstein blueberry soil showed a peak uptake of 13 pmol $g^{-1}$ $h^{-1}$ (P7L11). This does not seem to be just "20% stronger". Please reassess the figure and revise this statement.

> P8L2: "The OCS compensation points were found to be variable in close dependence on the soil water content."

This is not the case of the Waldstein blueberry soil. This exception needs to be acknowledged in the text.

> P8L14–15: "The values of $P_{OCS}$ were 2, 4, 7 and 30 pmol $g^{-1}$ $h^{-1}$ for Mainz soil, Waldstein soil with blueberry or young spruce understory, and Finland needle forest litter, respectively."

Which summary statistics are these numbers? Means or medians? Please specify.

> P8L20: "This might indicate involvement of multiple OCS uptake processes (see section 4.2)."

The observed OCS gross uptake vs. soil moisture patterns can be reproduced in model simulations even if there is just one OCS uptake process. See Sun et al. (2015) and

Ogée et al. (2016). This pattern can be explained as the result of two competing effects of soil moisture—inhibition on OCS diffusive transport and activation of microbial activity. In other words, it does not require the presence of multiple OCS uptake processes.

Discussion

The discussion sections, in general, are unnecessarily convoluted, cliché ridden, lack coherence, and fall short of offering novel insights. I struggle to find main points in a mixture of loosely connected statements. In each section, there is significant amount of superfluous or repeated information that needs to be pruned away to allow the point to get across.

4.1

> P8L24–25: "The OCS exchange from our laboratory measurements of the Finland litter layer soil is of the same magnitude as the field OCS exchange measurements performed by Sun et al. (2017) at the site where our samples were taken . . ."

The units are different (here, pmol $g^{-1}$ $h^{-1}$, and pmol $m^{-2}$ $s^{-1}$ in the cited study). How did you make the comparison?

> P8L29–30: ". . . the deposition velocities of our lab measurements were corrected based on the temperature optimum curve presented in Kesselmeier et al. (1999) by a factor of 0.852 (the ratio of OCS uptake at 15°C to OCS uptake at 20°C in Kesselmeier et al., 1999)."

This correction implies that the temperature dependence of the Finnish soil follows the same relationship in Kesselmeier et al. (1999), which was originally derived from an

agricultural soil, with a temperature optimum around 18°C. However, Sun et al. (2017) has shown that the temperature optimum of the Finnish soil exists at a much lower temperature. Therefore, the correction applied to the deposition velocity is not justified.

> P9–P10: "All three organic forest soil samples were almost exclusively OCS sinks, . . ."

The rest of this section seems purely speculative, theorizing with little support from the data, and strays further from the main point. The discussion is not a place for literature review. Even if there is an abundance of literature on autotrophic and heterotrophic microbes, OCS exchange data alone can neither verify nor falsify a particular microbial mechanism. Please reconsider what you intend to mean in this section.

4.2

> P11L2: "Wet soils have shown to tend towards emission of OCS"

'Wet soils' is not accurate, I suppose you mean 'water-saturated soils'.

> P11L8–9: "We suspect these processes to be connected to autotrophic organisms, as discussed in Section 4.4."

This statement is too speculative. First, there is no data supporting the presence of OCS consuming autotrophs in your samples. Second, heterotrophs, too, may consume OCS.

> P11L12–13: "This is in clear accordance with the assumption of a linear dependence of $U_{OCS}$ on the ambient OCS mixing ratio . . ."

This 'assumption' needs to be introduced in the Introduction first.

> P11L22–23: "Aside from that, only the experimental data obtained at 50 ppt and 1,000 ppt were used to calculate the compensation points."

This goes to the Methods, not the Discussion.

4.3

Rather than repeating what has been found in previous studies, more effort should be devoted to the interpretations of the compensation point results presented in this study.

4.4

The question why the '$k$-coefficient' varies with soil moisture and shows an optimum remains unanswered.

4.5

> P13L23: "Samples had been stored 5 to 9 months."

This information should be provided in the Methods.

> P13L23–24: "For both, there was uptake of OCS at about 12% gravimetric soil water content, which was reduced when the soil contained a higher or lower amount of water, gradually switching to emission of OCS at wet and very dry states as demonstrated in Figure 7."

Why not just say that the **optimal** uptake of OCS occurs at 12% SWC?

> P14L1–3: "Some changes occur both over time and are induced by two different ways of sample treatment (storing 'fresh' or air drying the sample before storage), illustrating the importance of consistent treatment and storage of samples that are meant to be compared to each other."

Is there evidence that the differences in soil moisture response patterns arise from the "two different ways of sample treatment"?

4.6

This whole section should either be removed or reduced to two sentences. The point boils down to the first sentence that soil OCS exchange is a small fraction of the ecosystem OCS exchange; the rest is merely a lengthy digression (P14L12–30). Not only does the latter half of the paragraph lack relevance, but it philosophizes over leaf area index without any support from empirical data. The purpose of this section needs to be reconsidered.

Conclusion

The Conclusion is poorly structured and reads like a first draft. Please find out the meaning and rewrite it.

Figure 1 & 2

Both figures show the patterns of OCS exchange versus soil moisture. What is the reason to split the data into two figures? Why not show the 500 ppt data and the

"Mainz dry soil" data in Figure 1?

Figure 3

Please consider adding error bars on compensation points. The uncertainty in OCS exchange should propagate to the compensation point.

Figure 4

1. The unit of $k_{OCS}$ is missing.

2. Net OCS flux is referred to as "net release" here, but "net exchange" in other parts. Please harmonize the terminology.

**Technical comments**

P1L12: "Carbonyl sulfide (OCS) is a chemically quite stable gas in the troposphere (lifetime ~2-6 years) and consequently some of it is transported up to the stratosphere where it contributes to the stratospheric sulfate layer."

This sentence is verbose. In the second half you may just say ". . . and is a precursor of the stratospheric sulfate aerosols."

There are updated estimates of OCS lifetime from Campbell et al. (2008): 1.5 to 3 years. (I assume that you got the 2–6 years estimate from Khalil et al., 1984.)

P1L13–14: "Due to the similarities in uptake mechanism between OCS and $CO_2$"

[Figure]

'uptake' → 'leaf uptake'

> P2L13: "Furthermore, changes of the sink strength of vegetation as a response to global change is a matter of discussion"

'is' → 'are'

> P3L4–5: "their dependences to environmental parameters . . ."

"their dependenc**e on** environmental parameters . . ."

> P7L23: "1000 ppm OCS"

I believe this should be "1,000 ppt OCS".

---

## Referee Comment (RC2) · Anonymous Referee #2 · 16 May 2018

The manuscript entitled "Carbonyl sulfide (OCS) exchange between soils and the atmosphere affected by soil moisture and compensation points" was submitted by Bunk, Yi, and coauthors for peer review to Biogeosciences. The authors present useful data on soils from four sites that help distinguish trends in simultaneous production and consumption processes that yield net OCS exchange in soils. The concept of the compensation point is used, which is not useful in the process-based sense, but can help identify soils that are more likely to be a source or sink to the atmosphere. While the results could be useful to a growing community of scientists studying OCS biosphere-atmosphere exchange processes, the paper and study design (as currently described) suffers from a number of significant flaws. Discussions of the implications regarding mi-

crobial drivers come across as forced and unsupported, the chemical measurements are underutilized, and many points require clarification. In this review, suggested improvements regarding organization, clarity, and pairing the scope main message to the scope of the study are given.

General comments:

Because the compensation point is a combination of multiple processes that each have different soil moisture dependencies, there is not a mechanistic reason that CP itself has a coherent moisture dependence. I would de-emphasize the CP relationship and focus more on was learned independently about P and U.

A sign convention should be adopted and net or gross uptake fluxes should be indicted with a '-' sign throughout the text and net or gross emissions with a '+' sign.

State the time duration over which the experiments were performed for a given soil from wet to dry. How can one distinguish between changes in the moisture dependency from changes in microbial communities and their CA over that time period? What implication could that have for kocs?

Improvements could be made to the organization of the introduction. For example, three drivers of OCS fluxes are mentioned first, two are explained in one paragraph and the third in another. In the introduction it should be stated what the reason is for comparing the fresh and dry soil.

Discussion paragraphs should be made more concise and shortened quite a bit. I would hesitate to claim 'good agreement' between the field and lab studies, but certainly the value are within the same range, suggesting that mechanistic conclusions from the lab can inform processes that may occur in the field.

A problem with the discussion point on litter (in 4.1) is that no other horizons were tested at the forest sites, making it difficult to determine whether it was the litter or other characteristics of forest soils that drive higher uptake rates. This should be acknowledged. Add the horizon or litter designation to Table 1 for each sample, as it is difficult to determine from the methods which horizons were used for each sample.

It would be helpful to clearly state how this study differs from previous results and what new information is provided

Specific comments:

P1L16: Not clear how the QCL is 'new', or why it is relevant. Appears multiple times.

P1L20: The values given are a range, which would be better terminology than "of the order", the meaning of which I don't understand.

P1L28: How are emissions excluded under lowest CPs?

P2L11: The last two sentences of this paragraph are difficult to understand. It's not clear if GPP is what is estimated from OCS or if OCS is used to estimate GPP. The introduction topics should be better developed to connect the last sentence to where the field currently stands in terms of OCS sink strength estimates.

P2L16: Explain how OCS:CO2 relationships are used to constrain the S cycle.

P2L22: Some orientation to which scale you are referring to would be helpful. Leaf, ecosystem, globe?

P2L29-P3L2: Not clear why it matters that early studies had artificially high OCS emissions because of measurement artifact unless you connect this explicitly to CPs for example.

P3L3: Is there a paper arguing the missing sinks early on that you could cite here?

P3L4: There is now a fair amount of literature on these drivers of OCS soil fluxes and mechanistic formulations representing them (see citations below). I'm sure this understanding can be tested in different soil, but is it fair to say the drivers are largely unknown? You cite additional papers to this point I the next few paragraphs, making

that statement seem a bit contradictory.

1. Ogée, J.; Sauze, J.; Kesselmeier, J.; Genty, B.; Van Diest, H.; Launois, T.; Wingate, L. A new mechanistic framework to predict OCS fluxes from soils. Biogeosciences 2016, 13, 2221–2240, doi:10.5194/bg-13-2221-2016.

2. Sun, W.; Maseyk, K.; Lett, C.; Seibt, U. A soil diffusion–reaction model for surface COS flux: COSSM v1. Geosci. Model Dev. 2015, 8, 3055–3070, doi:10.5194/gmd-8-3055-2015.

3. Whelan, M. E.; Hilton, T. W.; Berry, J. A.; Berkelhammer, M.; Desai, A. R.; Campbell, J. E. Carbonyl sulfide exchange in soils for better estimates of ecosystem carbon uptake. Atmos. Chem. Phys. 2016, 16, 3711–3726, doi:10.5194/acp-16-3711-2016.

4. Sun, W.; Maseyk, K.; Lett, C.; Seibt, U. Litter dominates surface fluxes of carbonyl sulfide in a Californian oak woodland. 2016, 438–450, doi:10.1002/2015JG003149.Received.

5. Kaisermann, A.; Ogée, J.; Sauze, J.; Wohl, S.; Jones, S. P.; Gutierrez, A.; Wingate, L. Disentangling the rates of carbonyl sulphide (COS) production and consumption and their dependency with soil properties across biomes and land use types. Atmos. Chem. Phys. Discuss. 2018, 1–27, doi:10.5194/acp-2017-1229.

P3L11: Cite also Whelan et al., 2016

P3L20: These are net sinks, but there can still be gross production in forest soils (Kasermann et al., 2018).

P3L23: Would be helpful to distinguish that you mean 'atmospheric or headspace' CO2 mixing ratios is what was controlled in that study (not pore-scale mixing ratios)

P5L15-27: The methods for calculating CP are circular and are not clearly explained. Please rewrite. Add additional equation notation to indicate the values determined where Eocs=0.

P5L28: I would clarify why the deposition velocity contains also the production term. This is for comparability to field measurements, correct?

P6L4: "Bourtsoukidis et al. (submitted)" is an unnecessary reference for this volumetric formulation. Please use a more suitable source. Define all terms in the equation (e.g., ti, ts) P6L18: If based on instrument precision, why would the resolution threshold vary by soil type? How is this different from your reported precisions L21, that have similar values?

P6L21: How did you determine the absolute mole fraction for the permeation source standard and for the certified gas mixtures? Which OCS source was used for which OCS mole fractions? What were their precisions? A supplemental figure showing the calibration curve should be shown to indicate how two of three types of calibration gases had excellent instrument response matching, but that the most high precision standard at atmospheric values was off by 7%.

P7L13: Is this gross uptake reported at the soil moisture where net uptake is highest? Clarify for your readers. 'Total uptake' is not clear. Please pick a set of terms for each meaning and stick to them throughout, such as: net exchange, net source, net sink, gross uptake, and gross production. Currently many terms are used in the text and figures (e.g., net release, exchange rate, total uptake, etc...). P7L15: The confidence intervals or standard deviation on these average emissions (not ranges) can be given or statistical tests performed to show if they are different from zero.

P7L20-30: This paragraph could be condensed significantly to make the main point (intermediate OCS mixing ratios give intermediate fluxes, which indicate compensation points).

P9L24-31: The argument here about higher heterotroph abundance and activity being related to higher Uocs is not well supported by the data in this study or any references listed. The cited literature is not referenced appropriately (e.g., Kato 2007 was looking at mycobacteria, which are heterotrophs). This is a poorly developed line of reasoning,

and I would suggest sticking to the chemical data in Table 2 to discuss possible drivers. I feel that this discussion point relies too strongly on submitted manuscripts that can't be accessed by reviewers, inaccurately referenced studies, and statements without any references. Instead, the discussion should be de-scoped to accurately integrate published results alongside the data from this work.

P10L14-22: The scope of this discussion point regarding production is much more in line with the paper (in contrast to aforementioned uptake sections) and seems quite appropriate.

P11L8: The role of diffusion limitation of substrate at high soil moisture levels should be discussed. Enzymes are likely active, but substrate limited in that case. This statement is not supported: "We suspect these processes to be connected to autotrophic organisms, as discussed in Section 4.4."

P11L10: The relationship between Eocs (and correspondingly Uocs shifted by the magnitude of Pocs) does not seem linear with OCS mole fraction, but instead increases significantly between 50 and 500 ppt and then less so from 500 to 1000 ppt. Please show a plot of the fits. What is a 'true trend'? If the relationship is not linear, errors in estimating U at E=0 may lead to errors in estimated kocs from two-point measurements. Are you confident this was not the case? This will bear on the discussion in 4.4

P12L8: Is there a specific heterotrophic CO2 fixation reference you can provide for the relation to OCS? The paper should focus on fluxes and CPs you observed without too much speculation on microbial metabolisms implicated. As stated by the authors, that activity was beyond the scope of the work.

---

## Referee Comment (RC3) · Anonymous Referee #3 · 16 May 2018

This is an interesting study that investigates how the net carbonyl sulfide (COS) exchange between a set of soils (forest and agricultural) and the atmosphere is composed of two opposing component fluxes (a gross uptake flux of COS and a gross production flux of COS) that are regulated differently in response to variations in soil water filled pore space. They revisit the compensation point method of Conrad, 1994 and Lehmann & Conrad, 1996 to obtain estimates of COS production from observations of the net soil COS exchange measured under two different atmospheric COS concentrations of 50 and 1000 ppt. These concentrations are much lower than those used in Lehmann & Conrad, 1996 and likely more applicable to concentrations observed in the field. Although many studies have measured and modeled from theory the response of

the net COS exchange to variations in soil moisture for a given soil and how differences in soil texture and bulk density play on the WFPS and subsequently the optimum net COS exchange (i.e. the maximum COS uptake rate measured), this is the first study to show that the COS production rate remains more or less constant over the entire range of %WFPS. Thus variations in the net COS exchange with WFPS are driven by the uptake component of the net COS flux. Between the different soils measured (3 spruce and 1 agricultural site) there were large differences in the magnitudes of the net COS flux and their component fluxes with largest fluxes for all components found in the Finnish forest and the lowest component fluxes observed in the agricultural soil in Germany. In addition a further experiment showed that after drying the agricultural soil for several months and re-humidifying, a very similar response of the net and component fluxes to % WFPS was observed, although no statistics were completed to test whether they were significantly different or not. In general there is some nice data in this study that is definitely worthy of publication.

**Major comments**

Throughout the manuscript there is very ambiguous application of terminology regarding the exact flux being presented. When the paper is expressly about partitioning the components of a net flux, one has to take care to be precise and state clearly which flux they are writing about. I thoroughly recommend that the authors go through the paper and clarify exactly what each flux is that they refer to, when they refer to it in the paper. Simply referring to COS exchange is too ambiguous, this paper must always refer to the net COS flux (EOCS), the COS emission rate or production rate (POCS) and the COS uptake rate (UOCS).

Furthermore, the partitioning approach taken in the current study does not completely isolate the two component gross fluxes, rather the uptake term measured at a constant temperature as presented by the authors is still regulated to some extent by diffusion (not strictly enzymatic uptake of COS) and the production term as presented still incorporates a COS deposition velocity (Vd0) that occurs even
when the COS concentration = 0. These details are developed in the Ogee et al., 2016 paper https://www.biogeosciences.net/13/2221/2016/bg-13-2221-2016.pdf and more relevantly to the current study in a recent publication in Atmospheric Chemistry and Physics Discussions by Kaisermann et al. https://www.atmos-chem-phys-discuss.net/acp-2017-1229/acp-2017-1229.pdf that demonstrates within the current methodological framework that a small additional analysis is required to obtain the gross COS production rate when COS concentration = 0 that must be solved iteratively see Eqs 2, 3 and 4 of Kaisermann et al.

Another important point that was expressed several times in the paper is that the shape of the net COS exchange response to WFPS is unknown but probably caused by changes in the activity of the enzyme CA. However, this is not strictly true as in the past few years the community has made considerable progress in explaining the response of the net COS flux to variations in soil temperature, soil moisture, soil texture and soil microbial biomass collectively in the papers of Kesselmeier et al., 1999; Van Diest & Kesselmeier, 2007; Ogee et al., 2016 and Sun et al., 2016. From these papers it has been shown that the observed optimum with soil moisture content observed in the present study and many times in the literature can be modelled extremely well and is mostly caused by changes in the diffusion of OCS within the soil matrix that reduces the potential hydrolysis rate at a given temperature, microbial biomass and COS concentration. This can occur over the typically short time frames of these experiments and thus the net COS flux and the gross COS uptake rate does not need to be driven by changes in the intrinsic enzyme activity or size of the microbial population. Thus the discussion needs to take this in to account and furthermore considered in the interpretation of the data presented in the results. Subsequently, differences in soil texture can probably explain most of the differences in the absolute %WFPS values where the optimum net COS flux is observed. Unfortunately, no data is provided in the manuscript about the differences in texture between the sites, this should be added.

As described above the data in this study are interesting however the presentation of
the results could definitely be improved and synthesised. I see no reason why figures 1 and 2 are not merged and a more synthetic analysis of the fresh vs dried/re-humidified soil results presented in a new Figure 2. This new figure could consist of 3 panels side by side. In panel (a) the authors would present the data from the 50ppt experiments, panel (b) the 500 ppt and panel (c) the 1000 ppt data. On the x-axis of each panel would be the net COS flux from the fresh soil against axis y the net COS flux from the same re-humidified dry soils. The authors could then colour the points by %WFPS (light blue = dry soils and dark blue = wet soils). Then they could also show the 1:1 line and do some regression statistics that way the audience can assess clearly and objectively the effect of the drying on the %WFPS response.

The authors also point out that the consumption rate coefficient (k) follows the same pattern with %WFPS as the net COS flux and the partitioned COS uptake rate (Uocs) and present the variability of k with %WFPS. However, U and k must vary with soil moisture and temperature ....etc as they are linearly related or even proportional if Vocs is always at the same COS concentration. Thus I would remove figure 5 and rewrite the discussion to address this point. Also the compensation point does not affect the COS exchange rate and thus the title of the manuscript should be corrected.

The authors state that the data from the Sun et al 2017 field study and this study are comparable and can be used to transfer the findings from the lab data to the field. However, the lab response to soil moisture content (green line) does not go through the middle of the points, but rather forms an upper envelope and there is no statistical test behind this statement. At the minimum the authors should calculate the mean deposition velocity for the relevant and comparable soil moisture values for their study and the Sun et al study and compare the means. Furthermore, I do not think it is appropriate to use the temperature optimum from the Mainz soil to make the field fluxes of the Finnish soil comparable. It has been demonstrated before that the net COS flux is strongly affected by the production rate and can cause a shift in the temperature optimum. As this study shows the Mainz and Finnish soils have extremely different

BGD
production rates I do not think this is the most appropriate way to reconcile the two data sets and facilitate comparison.

**Specific comments**

Page 3 of the introduction lacks a number of citations that describe the theoretical advances the research community has made in describing the response of net COS exchange to soil water, texture, soil temperature etc... there is also some internal contradiction within the text.

Finally the introduction does not present any hypotheses on why they might expect shifts in sources and sinks relative to their experimental manipulation with COS concentration and soil moisture content.

The characteristics of the soils provided are partially useful. It would be better to provide the physical characteristics such as bulk density or texture rather than nitrate and ammonium fluxes taken from some other studies and not conducted on the soils at the same time as when measured. These values could be misleading as inorganic N concentrations are turned over rapidly and vary with season and management.

Pg 4 line 10 what exactly can the author not exclude variability in over time? Was there no fixed protocol for the collection, storage and handling of the soils?

Pg 4 line 14 length of sample storage should be provided here and was it the same for each soil?

Section 2.4 and description of %WFPS protocol should come just after Section 2.1

Pg4 line 20 state the temperature of the soil here and how constant.

Pg 4 Section 2.2 how many sample replicates are measured for each soil at a time and what do the error bars refer to in the figures?

Section 2.2 How long between wetting and gas exchange started? How long is the measurement sequence? How long is the airstream sampled? How do you check for

BGD
steady-state? Information on when the soils were sampled would be useful e.g. time of year; before/after fetilisation?

Section 2.4 remove the citation Bourtsoukidis et al

Page 6 Line 24 is it the fluxes or absolute mixing ratios underestimated by 7%?

Page 7 line 12 I feel the last sentence of this paragraph is out of place and should appear later.

Section 3.2 really needs to be re-written. It is ambiguous, difficult to read and contains repetition. In places it is hard to work out what the authors are comparing.

Page 8 line 3 you should state that the soils had compensation points higher than the background atmosphere.

Section 3.3 again some repetition at the end of the paragraph.

Also the authors should point out that compensation points in themselves are not particularly useful as they are not intrinsic properties of a soil as their value will vary with temperature and COS concentration. They are only useful for establishing whether a soil is a source or sink of COS to the atmosphere.

Section 3.4 the figure 4 panel b production rate looks very strange. I am not sure why it has this appearance, but I am guessing maybe there is some interpolation being made between a limited number of measurements over the drying curve. It would be useful to explain what is going on here in the methods, results and the legend

Page 8 line 19-20 This statement as described above is redundant as is the figure 5 and is not linked to any of the things proposed in Section 4.2

Page 8 line 30 this correction may have some problems associated with it, if the authors insist on using it they should be more critical about why it is not ideal.

Page 9 Ln 3 the reason for the scatter is because many variables are changing at the

BGD
same time, please be a bit more critical.

Page 9 Ln 8 I would not hold this graph up as evidence that your study can simulate what is happening in the field.

Page 9 line 11-22 This discussion is a little imaginative and is not so relevant to the results. The arguments do not follow a clear logic.

Page 10 Line 11-13 I don't understand these statements. What exactly has it's activity reduced? What is the different uptake process and what evidence in your data do you have for this?

Page 10 line 22 What about abiotic processes?

Page 10 line 28-30 ambiguous statements about exchanges and observed values without being precise about what they are referring to. Which exchanges and values exactly?

Page 11 section 4.2 lines 7-15 I don't think any of these arguments are relevant to the results they are discussing. The authors do not appreciate the role of diffusion within the soil matrix and it's affect on the ability of soils to take up COS or not as WFPS changes. This should be explored first before jumping to the conclusion that autotrophic organisms have some role to play in explaining this pattern, especially as the authors have no evidence to support this hypothesis.

Section 4.2 last paragraph should be in the results section it is not discussion.

Page 12 line 3 also mention the other factors that will alter the compensation point.

Page 12 line 14-16 your experimental data does not support this statement about the two compensation points of Lehmann & Conrad please modify the sentence.

Page 13 Ln 2-19 this discussion again contains a lot of conjecture and fails to mention that the differences in texture between the spruce sites is probably important and should be accounted for before attributing differences in COS uptake rates to other Interactive comment

factors

Page 13 line 23 I think this is important info that should appear in the methods section.

Page 13 line 28-30 don't you think this is because fundamentally the soil texture did not change over the 6 years at the Mainz site and thus you observe a similar pattern when you wet and dry the soil 6 yrs later?

Page 13 line 18 LAI of 15? I don't believe this is possible

Section 4.6 I think this is a bit long and not so useful in the end I think it could be summarized in a sentence or two in the conclusion

Conclusions are currently based on conjecture and not the results. Statement 1 The experiment was not designed to test and cannot prove that COS is driven by the litter layer. Statement 2 There are no data presented about fungi in this paper so again this statement seems redundant. Statement 3 They do not prove that COS uptake is driven by different enzymatic processes. Statement 4 Is an introduced error of 1% really significant? Statement 5 No evidence in this study that the correlation coefficient k is linked to the presence or absence of auto- or heterotrophic organisms. Statement 6 I agree trying to understand compensation point variability without a model that accounts for how it varies with T, moisture and COS concentration is frustrating. We should use the theory and models that now exist to address this issue. Statement 7 they did not demonstrate statistically that the storage issues introduced significant differences in the fluxes and what level of uncertainty is introduced.

Merge Fig 1 and 2

Recommend a synthetic figure 2 with some statistical analysis.

Figure 3 should there not be some estimation of error on these points?

Figure 4 panel b looks weird also can you show that the inlet is constant during each of the experiments and that steady state was attained

BGD
Figure 5 redundant

Figure 6 not sure this is necessary either

Figure 7 not sure this figure is explicitly referred to in the text or necessary in the paper.

Table should state explicitly which nitrate and ammonium data are relevant to the gas exchange measurements taken during the actual present study experiment.

---

## Author Comment (AC1) · 28 Jun 2018

AUTHORS: We thank R1 for the critical and detailed review. Based on the input of R1, R2 and R3, the discussion section will undergo major changes and a restructuring.

General comments

REFEREE: The manuscript presents a laboratory study on two important drivers of soil OCS exchange—soil moisture and ambient OCS concentration. The authors collected four soil samples from the field, incubated them in the laboratory, and determined their OCS exchange patterns under varying conditions of soil moisture and OCS

concentrations. The experimental design covered the full range of relative soil moisture, from 0% to 100% with a fine step. This allowed the construction of well-defined soil moisture response curves of OCS exchange. By measuring OCS exchange under different ambient OCS concentrations, this study was also able to separate OCS gross uptake and production components from the net exchange and investigate their respective responses to soil moisture. However, the experimental design to derive compensation points was seriously flawed and unable to produce robust estimates of the compensation points. As the authors noted, they used measurements at 1,000 ppt and 50 ppt to derive the compensation points. This practice is essentially to fit a regression line with only two data points, and then extrapolate to somewhere up to 5,500 ppt! The uncertainty would likely be huge. Yet they present no uncertainty measures of this derived quantity in the figures or the text. That is not to say that the data shouldn't be published, but the authors should quantify the uncertainties and properly acknowledge the limitations of their method.

AUTHORS: We agree that the method used to derive compensation points in this study may be improved by investing much more measuring time and sample material. However we can rely on older reports about the linearity of OCS exchange between 50-1000 ppt in soils and plants. More data points to fit a regression would certainly be helpful. But it will further our understanding to present the current data set, also in view of the different soil types and chemical properties, and use it for discussion and development of new approaches. The limitations will definitely be addressed in a revised manuscript. However, our large data set provides a consistency over the whole range of the investigated soil water content, which covers a much wider range of soil moisture than in previous reports.

To address the uncertainties, we will add error estimations based on Gaussian error propagation for the net release (EOCS), uptake coefficient (kOCS), gross uptake (UOCS), gross production (POCS) and compensation points (CP). Of further importance is the question about linearity of the relationship between OCS concentration

and exchange rates at higher OCS mixing ratios. Earlier reports demonstrate good linearity up to approximately 1500 ppt OCS ambient concentration in measurements with soil (Kesselmeier et al., 1999). Investigations performed with lichens (Kuhn and Kesselmeier, 2000), which are lacking any outer protective cell layers and seem to behave similarly as soils, exhibit a comparable linearity relating exchange rates and OCS concentrations. In both cases the authors report a saturation effect at higher OCS concentrations, which leads to an increasing uncertainty. The reasons are not investigated but can be understood as metabolic saturation (enzyme saturation) or different/additional biotic processes influencing the rate of OCS exchange. These uncertainties limit the reliability of the compensation points calculated, especially if those are >1500ppt or at soil moistures above 60 % WFPSlab. However, despite this uncertainty, the CPs show a clear and consistent trend that cannot be disregarded. It can clearly be seen that the uptake coefficient (slope of the regression) first increases and then decreases again with decreasing soil moisture. Setting the reliability of the absolute values aside, such coherent trends contain information that can be explored. Additionally, the calculated CPs have reasonable overlap with older data. An important aspect of the paper is the observation of CPs over soil moisture which has not been reported before. Even with relatively high uncertainties this data is worth to pursue and to plan future experiments accordingly. For the current revision we propose to test the significance of all trends in EOCS, kOCS and POCS by using the first data point of each measurement as control and perform a t-test to see if the next value is statistically different from the previous one. It would highlight the idea of monitoring a significant trend in OCS over drying out.

REFEREE: The manuscript identified the compensation point as a driver of OCS exchange, as the title and some section titles implied. But a critical examination of the mechanisms governing OCS uptake and production processes suggests that this is a misinterpretation. Compensation point manifests the dynamic balance between uptake and production. It is not an intrinsic property of the soil, but a variable that depends on soil moisture, as the authors have demonstrated in Fig. 3. Therefore, it is inaccurate to

say that OCS exchange is 'affected' by the compensation point. The manuscript suffers from other flaws. The equations (Eqs 1 and 2) used to calculate the OCS exchange apply only to the steady state condition, which can be shown mathematically by solving a mass balance equation for chamber gas exchange. Yet, they did not assure the readers of the validity of the steady state assumption used in their measurements.

AUTHORS: See specific comments

REFEREE: In addition, the interpretations of data are insufficient, and sometimes superficial. I would suggest the authors to exploit their dataset for new understanding rather than confirmatory interpretations. The discussion is poorly organized and lacks novel insights. This part has to be rewritten for clarity and coherence. As a suggestion, please figure out the main point of each section and organize the paragraphs in a streamlined way that helps convey the main point and impart understanding to the readers. Detailed comments on specific issues are listed below. AUTHORS: The revised manuscript will be rewritten.

Specific comments

REFEREE: Abstract; P1L24–26: "The OCS compensation points (CPs) were highly dependent on soil water content and extended over a wide range of 130 ppt to 1600 ppt for the forest soils and 450 ppt to 5500 ppt for the agricultural soil." If OCS exchange was measured only at "50, 500, and 1000 ppt" (P1L18), how did you obtain a compensation point of 5500 ppt for a certain sample? This has not been explained in the manuscript. Presumably, the only way would be to extrapolate a linear relationship between the net exchange and the concentration. The compensation points derived this way will inevitably have large uncertainties, since there were only three (and sometimes just two!) data points to fit a line. Please provide the uncertainty measures and address the limitations of this method.

AUTHORS: As already given above, we agree that the uncertainty and limitations of our method for determining the CPs should be addressed. We propose to mention that

the method is limited and naming the magnitude of uncertainty here. Section 2.5 or a new section 2.6 seem to be a better place for a detailed discussion of this matter. We propose not go into too much detail in the abstract but handle it in the corresponding sections instead.

REFEREE: Introduction, P2L6: "In view of the potential role of OCS, . . ." What kind of 'potential role'? Please clarify.

AUTHORS: In this context the potential role in radiative forcing/cooling.

REFEREE: P2L2–8: "Carbonyl sulfide (OCS) is . . . in the troposphere (Brühl et al., 2012)." I think the first paragraph of the Introduction can be shortened significantly without losing the sense.

AUTHORS: We see no need to further shorten this already compressed overview.

REFEREE: P2L16: "in both the carbon and sulfur cycles". The 'sulfur cycle' is not a major concern of the studies cited here. I suggest removing it.

AUTHORS: We disagree. The carbon and sulfur cycle are overlapping in all these discussions. A new GPP estimation based on the sulfur exchange and the role and relations of sources and sinks for the carbon and sulfur cycle (here OCS) is a hot discussion and this approach can be exploited from both sides.

REFEREE: P2L15–26: "The relationship between concentrations of OCS and gross primary production (GPP) . . ." This paragraph is an elaboration on the use of OCS as a tracer for GPP. While broadly speaking this is a motive for many recent studies on soil OCS exchange, as plant uptake of OCS is not the topic of this study, having such a level of details in the Introduction seems excessive. I suggest cutting this paragraph down to three sentences or so.

AUTHORS: As the referee points out, the use as a GPP tracer is one of the main motivations of OCS research. As such, dedicating 11 lines in the introduction to pointing out the background does not seem excessive to us, as one of the points of an introduction is providing the context to the field of research. Readers that are more familiar with the topic and not interested in rereading can easily skip the paragraph. The role of forest soils is of significant interest in using the net OCS exchange as a proxy for forest GPP.

REFEREE: P3L3–4: "Although the understanding of soils as a major sink helps to explain the 'missing sinks' for OCS, soil uptake still shows a wide scatter among different environments. "Where does the "missing sink" come from, all of a sudden? Please explain.

AUTHORS: This was a translation mistake. We apologize. This part will be rewritten to: "Although the understanding of soils as a major sink helps to explain the general role, soil uptake still shows a wide scatter among different environments."

REFEREE: P3L4–5: "Also, the drivers of soil OCS fluxes and their dependences to environmental parameters (such as soil moisture, soil temperature, and OCS mixing ratio) are still largely unknown." This is not an accurate statement. Actually, you have cited quite a few studies in the next few paragraphs to show that the drivers are not 'largely unknown.'

AUTHORS: That is true. We propose to remove this sentence. Instead, after the list of known drivers with their citation we propose to add: "Unfortunately, the way by which these drivers influence the soil-atmosphere exchange and the quantity of their impact is not fully understood and quantified yet." A reference to the different temperature relationships (Q-10 values) for Pocs and kOCS that have recently been identified by Kaisermann et al., 2018 (ACP, Discussion Paper) will also be included.

REFEREE: P3L5–6: "Uncovering the mechanisms for soil OCS fluxes would allow soil atmospheric OCS exchange to be estimated on broader spatial scales." I think that Berry et al. (2013) and Launois et al. (2015) have already estimated soil–atmosphere OCS exchange 'on broader spatial scales'. Please clarify what you mean by this statement.

AUTHORS: The current global estimates are a hot matter of discussion with large questions for oceans and soils regarding sink/source strengths. The current publications are by no way the final answer! We propose to rewrite to: "Uncovering the mechanisms for soil OCS fluxes would allow soil atmospheric OCS exchange to be estimated on a reliable basis to increase the reliability of OCS exchange estimations from local and regional to broader spatial scales."

REFEREE: P3L12–17: "Furthermore, there is a strong evidence that the OCS exchange between soil and the atmosphere is dependent on the ambient OCS mixing ratio . . .". What's missing from the introduction of the compensation point is the fundamental cause of it: OCS uptake is a pseudo first-order reaction and thus depends linearly on the ambient OCS concentration, whereas OCS production is independent of OCS concentration (Conrad, 1994).

AUTHORS: We propose to move the paragraph about compensation point calculation from section 2.3 to here (slightly modified), as you suggested later on.

REFEREE: P3L24: "To deepen our understanding of source and sink characteristics ...". At the end of the Introduction it is still not clear to the readers what specific research questions this study aims to address. The authors should consider formulating research questions or hypotheses to better orient the readers.

AUTHORS: We agree that adding research questions to the introductions will be an improvement. We propose to add the following questions:

(1)What is the relationship of net OCS release (EOCS), gross OCS consumption (UOCS), gross OCS Production (POCS) and the OCS compensation points to soil moisture and what are possible mechanisms behind this relationship?

(2)How do chemical properties, physical properties and origin (topsoil samples such as Mainz soil and Finland soil vs. organic layer soils from Waldstein sites) influence the net OCS release (EOCS), gross OCS consumption (UOCS), gross OCS Production

(POCS) response to different soil moistures.

(3)Are there differences in the OCS net release and other properties when a sample has been stored "field fresh" or dried before measurement?

(4)Are our laboratory measurements comparable to reported field measurements at the sampling site?

REFEREE: Material and methods, P4L3: "Soil samples were collected from four sites . . .", P4L9–10: "We tried to use the same method to collect all soil samples, but we cannot exclude the variability over time." Can you provide information on when each sample was collected?

AUTHORS: The sampling was 2012, 2012 and 2014 for the Finland soil, the Waldstein soils and the Mainz soil, respectively. We propose adding (sample collected 201X) with the proper year to each soil as they are described in section 2.1

REFEREE: P4L10–11: "Fresh subsamples of agricultural soil in Mainz were oven dried (at 40°C) for comparison." Please specify how long they had been dried for in the oven. Typically, using the gravimetric method to determine soil moisture would require samples to be oven dried at 105° C for 48 to 72 hours, but of course that temperature would not be desirable because microbial communities could be destroyed. I assume that the choice of 40° C had something to do with this concern, but at this temperature, how did you ensure that the samples were dried completely and the maximum soil moisture used in Eqn (6) was not biased by any remaining water?

AUTHORS: The drying at 40°C described here has nothing to do with the determination of maximum soil moisture. It is just to prepare storage of active soils. We wanted to compare to discern between storage techniques. For the determination of the maximum soil moisture and soil dry weight used in each measurement, all samples were dried at the end of a measurement at 105° C for 48h. To avoid confusion, we propose to include this information both here and in section 2.4.

REFEREE: P4L16 and P4L26: "deionized water (R 18.2 MΩ)". The unit of DI water resistivity is MΩ_cm, not ΩM.

AUTHORS: Yes. This will be corrected.

REFEREE: P5L21–23: "This is based on the linear relationship between OCS uptake and OCS mixing ratio shown by Kesselmeier et al. (1999) and the assumption that the ambient OCS mixing ratio does not influence OCS production(see 4.1)." This sentence explains the theoretical basis of the 'compensation point' and should better be placed in the Introduction.

AUTHORS: We agree that this should be part of the introduction.

REFEREE: P5L9, Eqn (1): EOCS=ΔOCS×Q/m_soil

This equation works only under a steady state condition. How did you ensure that OCS concentration in the chamber headspace reached a steady state? Please provide relevant details and reasoning.

AUTHORS: That is not true. Using this equation for dynamic equilibria is a long standing and accepted practice. See Breuninger et al. (2012) for detailed explanations. Breuninger et al. explain leaf chambers, but the same principle applies to soil chambers. See R. Oswald et al. (2013) or similar publications for examples of eq1 being used for dynamic chamber measurements. In short: In a well-mixed chamber (ensured by the activity of a fan in the headspace of each cuvette in our setup) the concentration of a trace gas is determined by the concentration in the flushing gas, the rate of flushing and the release or uptake by the soil. The flushing rate is constant and monitored. Factors like soil moisture are very slow changing in relation to the measurement period averaged for calculation by eq1. Chamber effects are accounted for by using a soil free identical chamber for reference. While true steady state conditions can only be archived with static chambers, a dynamic equilibrium was archived in flushed chambers that fulfills all conditions required to employ eq1. This dynamic

equilibrium was ensured as described below and can/will be involved in the revised paper: Switching the analyzer inlet from one chamber to the next of course courses some disturbances in the chamber and also the inlet tube needed flushing. We did extensive pre-measurements by observing measured chamber concentration with chambers that were empty or loaded with soil samples, waiting for steady concentrations to be measured. With the length of inlet tube and flushing rate used, stable concentrations were always observed after less than six minutes. Based on these empirical determinations, the time each chamber was connected to the analyzer was set to ten minutes. Of these ten minutes, the first 7 minutes were discarded to allow for proper equilibration within the chamber and at the same time allowing proper flushing time for the analyzer and analyzer inlet tube. The following 2.5 minutes were averaged for 30 seconds intervals and used for exchange calculations. The last 30 seconds of each chamber cycle was discarded again to eliminate any chance of accidental overlap.

REFEREE: Results, P7L6–7: "OCS uptake at medium soil moisture was 20% stronger for the Waldstein soil with its young spruce understory than for the one with a blueberry understory." What range is the "medium soil moisture"? Be quantitative. Waldstein spruce soil showed a peak uptake of 23 pmol $g-1$ $h-1$ and Waldstein blueberry soil showed a peak uptake of 13 pmol $g-1$ $h-1$ (P7L11). This does not seem to be just "20% stronger". Please reassess the figure and revise this statement.

AUTHORS: The number will be revised to the proper percentage. Medium soil moisture will be replaced by 35 to 50% WFPSlab.

REFEREE: P8L2: "The OCS compensation points were found to be variable in close dependence on the soil water content." This is not the case of the Waldstein blueberry soil. This exception needs to be acknowledged in the text.

AUTHORS: That is not entirely true. The dependence is less pronounced but some minima exist at 25 % WFPSlab and 90+ % WFPSlab. This is caused by the overall weaker reaction of UOCS to soil moisture in this soil. We will revise to acknowledge

the relationship being weaker than with the other soils.

REFEREE: P8L14–15: "The values of POCS were 2, 4, 7 and 30 pmol g−1 h−1 for Mainz soil, Waldstein soil with blueberry or young spruce understory, and Finland needle forest litter, respectively." Which summary statistics are these numbers? Means or medians? Please specify.

AUTHORS: Medians, rounded to full pmol. The median for the Finland soil will be revised to 27 pmol g-1 h-1. The text will be revised to "median POCS".

REFEREE: P8L20: "This might indicate involvement of multiple OCS uptake processes (see section 4.2)." The observed OCS gross uptake vs. soil moisture patterns can be reproduced in model simulations even if there is just one OCS uptake process. See Sun et al. (2015) and Ogée et al. (2016). This pattern can be explained as the result of two competing effects of soil moistureâ̆inhibition on OCS diffusive transport and activation of microbial activity. In other words, it does not require the presence of multiple OCS uptake processes.

AUTHORS: Though we generally agree with the referee, that microbial activity and diffusion can model a significant part of the exchange, we disagree when looking closer. There is a pattern of maximum OCS uptake under moderate soil moisture for Mainz soil and Finland needles and this is highly different to the other soils and must be discussed. Furthermore, the chemical quality may be of further relevance with special respect to the ammonium content. Additionally, we should regard a mix of enzymatic reactions (CA, Rubisco, PEP-Co and others?) and a mix of organism.

REFEREE: Discussion, P8L24–25: "The OCS exchange from our laboratory measurements of the Finland litter layer soil is of the same magnitude as the field OCS exchange measurements performed by Sun et al. (2017) at the site where our samples were taken . . .". The units are different (here, pmol g−1 h−1, and pmol m−2 s−1 in the cited study). How did you make the comparison?

AUTHORS: As we have both the soil mass and the chamber diameter we can easily calculate the OCS exchange of our samples either in pmol g-1 h-1 and in pmol m-2 s-1, as mentioned in section 2.3. It is only a question about to which reference value (surface area or sample weight) the concentration differential is referred. For consistence with our other publications we chose to show our data in pmol g-1 h-1 but for the comparison with Sun et al. we referred to the exchange data calculated according to Eq2. We will make sure to clarify this again in this section.

REFEREE: P8L29–30: ". . . the deposition velocities of our lab measurements were corrected based on the temperature optimum curve presented in Kesselmeier et al. (1999) by a factor of 0.852 (the ratio of OCS uptake at 15° C to OCS uptake at 20° C in Kesselmeier et al., 1999)." This correction implies that the temperature dependence of the Finnish soil follows the same relationship in Kesselmeier et al. (1999), which was originally derived from an agricultural soil, with a temperature optimum around 18°C. However, Sun et al. (2017) has shown that the temperature optimum of the Finnish soil exists at a much lower temperature. Therefore, the correction applied to the deposition velocity is not justified.

AUTHORS: Sun et al. explicitly state they could not find a temperature optimum (mainly because they could only measure a small range of temperatures, as per their own statement). See one representative quotation from Sun et al., 2017 at the end of this answer below. Of course one could take that as an argument for not using a temperature correction in itself. But we still think that using the Kesselmeier et al. 1999 temperature curve is the best approximation (though not optimal) we can get. Another approach would be to follow the Kaisermann et al., 2018. They found 1.23 for kOCS Q10 value for 10° C and a large set of different soils. So for 5° C it should be 0.615. Therefore, we can either make no correction at all and ascribing the higher uptake in our measurements to the higher temperature (pointing out the small t-range in the Sun et al. measurements) or use the Kesselmeier et al. 1999 temperature curve as a correction to the best possible approximation and discuss in the text that there

probably is some shift because of different temperature optima. We prefer the latter procedure. Statement in Sun et al 2017, p8l30: "However, a temperature optimum for COS uptake cannot be identified for our site. This is not surprising given that 90% of the data were measured at humus layer temperature in the range of 8.3–16.4°C (below 30 the optimum temperature of ca. 20° C, for example, observed by Kesselmeier et al., 1999), and that temperature and moisture co-vary in natural conditions"

REFEREE: P9–P10: "All three organic forest soil samples were almost exclusively OCS sinks, . . .". The rest of this section seems purely speculative, theorizing with little support from the data, and strays further from the main point. The discussion is not a place for literature review. Even if there is an abundance of literature on autotrophic and heterotrophic microbes, OCS exchange data alone can neither verify nor falsify a particular microbial mechanism. Please reconsider what you intend to mean in this section.

AUTHORS: based on reviewer input and reconsiderations we have decided to remove the discussion about heterotrophs and autotrophs as well as strongly reducing most discussion of microbial communities. Instead, other options as physical and chemical soil properties shall be included in the discussion of the revised version. Especially a topic of diffusion limitation by high soil moisture and a brief discussion of the possible role of Ammonium oxidizing organisms, as there are significant differences in Ammonium content in the examined soils that coincide with curve shapes of UOCS and kOCS, will be included.

REFEREE: P11L2: "Wet soils have shown to tend towards emission of OCS" 'Wet soils' is not accurate, I suppose you mean 'water-saturated soils'.

AUTHORS: We agree that 'water-saturated soils' is much more accurate and will adapt this wording.

REFEREE: P11L8–9: "We suspect these processes to be connected to autotrophic organisms, as discussed in Section 4.4." This statement is too speculative. First, there

is no data supporting the presence of OCS consuming autotrophs in your samples. Second, heterotrophs, too, may consume OCS.

AUTHORS: We have decided to remove all discussion about autotrophs and heterotrophs, as stated above. Also, the section can be shortened as it is covered in our follow up study already under review. We propose a merger of the current sections 4.2 and 4.3 (as kOCS and EOCS are necessarily tightly related) by compressing the points discussed therein and adding additional points of view (chemical and physical soil properties, diffusion limitation). One aspect that should be highlighted is the role of Ammonium, as there are major differences in the Ammonium content of the soils in this study.

REFEREE: P11L12–13: "This is in clear accordance with the assumption of a linear dependence of UOCS on the ambient OCS mixing ratio . . ." This 'assumption' needs to be introduced in the Introduction first. P11L22–23: "Aside from that, only the experimental data obtained at 50 ppt and 1000 ppt were used to calculate the compensation points." This goes to the Methods, not the Discussion.

AUTHORS: We agree this belongs into the methods section and should be added there. However, when discussing the results, it makes sense to mention the fact here, too.

REFEREE: The question why the 'k-coefficient' varies with soil moisture and shows an optimum remains unanswered.

AUTHORS: The k-coefficient is dependent on EOCS. Since this varies with soil moisture, so will kOCS. kOCS was used here because it is better suited to illustrate how soil moisture affects

REFEREE: P13L23: "Samples had been stored 5 to 9 months." This information should be provided in the Methods.

AUTHORS: We will add the information to section 2.1

REFEREE: P13L23–24: "For both, there was uptake of OCS at about 12% gravimetric soil water content, which was reduced when the soil contained a higher or lower amount of water, gradually switching to emission of OCS at wet and very dry states as demonstrated in Figure 7." Why not just say that the optimal uptake of OCS occurs at 12% SWC?

AUTHORS: Because that would leave out a lot of relevant information. P14L1–3: "Some changes occur both over time and are induced by two different ways of sample treatment (storing 'fresh' or air drying the sample before storage), illustrating the importance of consistent treatment and storage of samples that are meant to be compared to each other." Is there evidence that the differences in soil moisture response patterns arise from the "two different ways of sample treatment"?

AUTHORS: Both samples are from the same site and were collected together (actually one sample was divided after collection and homogenization into 2 subsamples). They underwent the same experimental procedure at the same time. They were stored under the same conditions (dark, 4 °C.) The only difference is their treatment before storage. So while it cannot be excluded that the differences were caused by random fluctuations, the pre-storage preparation is the only difference in the two samples which may explain our results. Prominently, this pre-storage treatment led to an increase of the ammonium content of the soil. According to Thion and Prosser (2014), Ammonium oxidizing bacteria (AOB) get reactivated after drying very fast and utilize the ammonium enriched during the drying.

REFEREE: This whole section should either be removed or reduced to two sentences. The point boils down to the first sentence that soil OCS exchange is a small fraction of the ecosystem OCS exchange; the rest is merely a lengthy digression (P14L12–30). Not only does the latter half of the paragraph lack relevance, but it philosophizes over leaf area index without any support from empirical data. The purpose of this section needs to be reconsidered.

AUTHORS: We agree and propose scrapping section 4.6, moving the main message (OCS exchange from that soil being only a small fraction of the ecosystem exchange under normal conditions) in a much reduced form to another section.

REFEREE: The Conclusion is poorly structured and reads like a first draft. Please find out the meaning and rewrite it.

AUTHORS: The conclusions will be rewritten, following the research questions from the introduction.

REFEREE: Figure 1 & 2, Both figures show the patterns of OCS exchange versus soil moisture. What is the reason to split the data into two figures? Why not show the 500 ppt data and the "Mainz dry soil" data in Figure 1?

AUTHORS: We agree and will merge Figure 1 and Figure 2.

REFEREE: Figure 3, Please consider adding error bars on compensation points. The uncertainty in OCS exchange should propagate to the compensation point.

AUTHORS: We agree and will add error bars.

REFEREE: Figure 4, 1. The unit of kOCS is missing. 2. Net OCS flux is referred to as "net release" here, but "net exchange" in other parts. Please harmonize the terminology.

AUTHORS : 1. The unit is mol g-1 h-1 and will be added in all relevant places.

AUTHORS : 2. Agreed. EOCS shall be "net release", UOCS shall be gross uptake and POCS shall be gross production and in all instances only these terms will be used. We are using the term release because "flux" is defined as in pmol m-2 h-1, while a release is in pmol g-1 h-1.

Technical comments

REFEREE: P1L12: "Carbonyl sulfide (OCS) is a chemically quite stable gas in the

troposphere (lifetime ∼2-6 years) and consequently some of it is transported up to the stratosphere where it contributes to the stratospheric sulfate layer." This sentence is verbose. In the second half you may just say ". . . and is a precursor of the stratospheric sulfate aerosols."

AUTHORS: Both are acceptable.

REFEREE: There are updated estimates of OCS lifetime from Campbell et al. (2008): 1.5 to 3 years. (I assume that you got the 2–6 years estimate from Khalil et al., 1984.)

AUTHORS: Agreed, we will use the updated estimation

REFEREE: P1L13–14: "Due to the similarities in uptake mechanism between OCS and CO2" 'uptake' → 'leaf uptake'

AUTHORS: Yes, thank you. It is of course better to be specific.

REFEREE: P2L13: "Furthermore, changes of the sink strength of vegetation as a response to global change is a matter of discussion": 'is' → 'are'

AUTHORS: Yes, thank you. We will correct this.

REFEREE:P3L4–5: "their dependences to environmental parameters . . ." "their dependence on environmental parameters . . ."

AUTHORS: Yes, thank you. We will correct this.

REFEREE: P7L23: "1000 ppm OCS" I believe this should be "1000 ppt OCS".

AUTHORS: Yes, thank you. This was a typo.

Referenced literature:

Breuninger, C., Oswald, R., Kesselmeier, J., and Meixner, F. X. (2012) The dynamic chamber method: trace gas exchange fluxes (NO, NO2, O3) between plants and the atmosphere in the laboratory and in the field, Atmos. Meas. Tech., 5, 955-989, doi:10.5194/amt-5-955-2012.

R. Oswald et al. (2013) HONO Emissions from Soil Bacteria as a Major Source of Atmospheric Reactive Nitrogen. Science 341, 1233, DOI: 10.1126/science.1242266.

Kaisermann, A., Ogée, J., Sauze, J., Wohl, S., Jones, S. P., Gutierrez, A., and Wingate, L.: Disentangling the rates of carbonyl sulphide (COS) production and consumption and their dependency with soil properties across biomes and land use types, Atmos. Chem. Phys. Discuss., https://doi.org/10.5194/acp-2017-1229, in review, 2018

Kuhn, U. and Kesselmeier, J. (2000) Environmental variables controlling the uptake of carbonyl sulfide by lichens. J. Geophys. Res. 105 (22), 26783-26792.

Thion, C., & Prosser, J. I. (2014). Differential response of nonadapted ammonia-oxidising archaea and bacteria to drying–rewetting stress. FEMS microbiology ecology, 90(2), 380-389.

---

## Author Comment (AC2) · 28 Jun 2018

REFEREE: The manuscript entitled "Carbonyl sulfide (OCS) exchange between soils and the atmosphere affected by soil moisture and compensation points" was submitted by Bunk, Yi, and coauthors for peer review to Biogeosciences. The authors present useful data on soils from four sites that help distinguish trends in simultaneous production and consumption processes that yield net OCS exchange in soils. The concept of the compensation point is used, which is not useful in the process-based sense, but can help identify soils that are more likely to be a source or sink to the atmosphere. While the results could be useful to a growing community of scientists study-

ing OCS biosphere-atmosphere exchange processes, the paper and study design (as currently described) suffers from a number of significant flaws. Discussions of the implications regarding microbial drivers come across as forced and unsupported, the chemical measurements are underutilized, and many points require clarification. In this review, suggested improvements regarding organization, clarity, and pairing the scope main message to the scope of the study are given.

AUTHORS: We thank the referee for the work he invested to improve our manuscript.

REFEREE: General comments: Because the compensation point is a combination of multiple processes that each have different soil moisture dependencies, there is not a mechanistic reason that CP itself has a coherent moisture dependence. I would de-emphasize the CP relationship and focus more on was learned independently about P and U.

AUTHORS: That depends on the view. If we look at the exchange with a view of an atmospheric scientist, the exchange is affected by a compensation point. At higher concentration it switches to an uptake and at lower concentration OCS is emitted. It is not simply variable. But we agree to rewrite to: OCS exchange exhibits a CP. Additional consideration how soil properties may affect UOCS will also be included. We agree with the referee that these considerations will represent a very useful complementation for the discussion of the compensation point. One main point of interest will be the difference in Ammonium content and how it may be linked to OCS uptake.

REFEREE: A sign convention should be adopted and net or gross uptake fluxes should be indicted with a '-' sign throughout the text and net or gross emissions with a '+' sign. AUTHORS: This is a tackling a basic discussion between meteorologists and physiologists. Do we need a positive or negative sign for example in case of photosynthetic uptake of CO2? However, we fully agree that a sign and a term convention can be very helpful to foster understanding of our data. We will check our revised manuscript carefully.

REFEREE: State the time duration over which the experiments were performed for a given soil from wet to dry. How can one distinguish between changes in the moisture dependency from changes in microbial communities and their CA over that time period? What implication could that have for kocs?

AUTHORS: This point goes back to the beginning of the review, where the referee stated that the concept of CP is not useful in the process-based sense. We agree to that statement, but want to add, that indeed the disentangling of POCS and kOCS contain valuable information which allows interpretation with respect towards microbial metabolism. In a recent modelling study (Ogee et al., 2016) kOCS was modelled over various environmental conditions (eq. 11b) using k_uncat, and a temperature dependency for kcat/KM (only one value is known in literature for certain pH and T: 2.39 s-1 $\mu$M-1, which is used to parameterize the model for various pH and T), CA concentration [CA] (was kept constant over dry-out) expressed as fCA, the CA enhancement factor. Thus assuming that in our short incubations no major growth occurred, the difference in kOCS for Mainz soil/Finland needle soil versus Waldstein blueberry/Waldstein spruce could be explained by either different isoforms of CA (Sauze et al. 2017) having different kcat/KM values (or assuming other enzymes such as RubisCO (Whelan et al., 2018) being involved, or that the enzyme concentration of CA (or others such as RubisCO) is not constant during drying-out as assumed by Ogee et al., 2016. Finally, we want to point out that the role of CA, RubisCO and PepCO in plants and soil is still not fully understood. Which effect was the dominant one, would need simultaneous molecular/enzyme analysis which was out of scope for this work.

REFEREE: Improvements could be made to the organization of the introduction. For example, three drivers of OCS fluxes are mentioned first, two are explained in one paragraph and the third in another. In the introduction it should be stated what the reason is for comparing the fresh and dry soil. Discussion paragraphs should be made more concise and shortened quite a bit. I would hesitate to claim 'good agreement' between the field and lab studies, but certainly the value are within the same range,

suggesting that mechanistic conclusions from the lab can inform processes that may occur in the field. A problem with the discussion point on litter (in 4.1) is that no other horizons were tested at the forest sites, making it difficult to determine whether it was the litter or other characteristics of forest soils that drive higher uptake rates. This should be acknowledged. Add the horizon or litter designation to Table 1 for each sample, as it is difficult to determine from the methods which horizons were used for each sample. It would be helpful to clearly state how this study differs from previous results and what new information is provided

AUTHORS: We will rewrite and restructure to incorporate the proposals for an improvement of the manuscript. We will clarify and summarize what we regard as new.

REFEREE: Specific comments, P1L16: Not clear how the QCL is 'new', or why it is relevant. Appears multiple times.

AUTHORS: Its new in the sense that cavity ring down and cavity output laser instruments that can measure OCS concentrations online at normal atmospheric concentrations are a fairly new development (about 6 years now). Measurements as in this study would not have been possible with the older techniques, which required preconcentration via cryo-trapping and could only produce one data point every 15 minutes or so (compared to a measurement rate of 1Hz and more).

REFEREE: P1L20: The values given are a range, which would be better terminology than "of the order", the meaning of which I don't understand.

AUTHORS: we agree and will replace "of the order" with "in the range"

REFEREE: P1L28: How are emissions excluded under lowest CPs?

AUTHORS: These CP are far below normal atmospheric concentrations, so net emission could occur only under highly unusual conditions.

REFEREE: P2L11: The last two sentences of this paragraph are difficult to understand. It's not clear if GPP is what is estimated from OCS or if OCS is used to estimate GPP.

The introduction topics should be better developed to connect the last sentence to where the field currently stands in terms of OCS sink strength estimates.

AUTHORS: Both is possible and done according to the literature. We will clarify this a bit more.

REFEREE: P2L16: Explain how OCS:CO2 relationships are used to constrain the S cycle.

AUTHORS: We agree this is a somewhat incomplete statement. The (atmosphere-biosphere) S Cycle is touched in the cited papers but OCS:CO2 relationship is not directly exploited that way. But the approach of measuring OCS fluxes and deduct the CO2 flux into vegetation from it can be reversed to estimate OCS uptake from CO2 uptake. OCS and CO2 are metabolized by the same enzymatic pathways. Hence, CO2 assimilation can be regarded as a proxy for OCS deposition and vice versa. This view opens a window to the Sulphur cycle. The impact of OCS uptake on plant needs for sulfur on the other hand is most likely very low, at least for plants with roots which may take up sulfur compounds from the soil. Nevertheless, OCS which is metabolized to H2S and fed into the plant sulfur cycle may be regarded to contribute, an assumption which has been discussed earlier for lichens with no connection to soil (Kuhn et al., 2000). Therefore, we think that the OCS:CO2 relationship may be also exploited to contribute to our understanding of the sulfur cycle. This has been demonstrated using numbers of carbon uptake to estimate global OCS fluxes (see Sandoval-Soto et al., 2005).

REFEREE: P2L22: Some orientation to which scale you are referring to would be helpful. Leaf, ecosystem, globe?

AUTHORS: This sentence is a continuation of the statement of the sentence before, referring to leaf uptake. However, it is likewise true on an ecosystem scale. On that scale the exchange behavior of soils complicates the picture which is addressed later on. Soil exchange is considered so low in comparison to vegetation exchange that it

is often considered negligible in these considerations. However, this is an uncertain assumption and may not be true under all conditions.

REFEREE: P2L29-P3L2: Not clear why it matters that early studies had artificially high OCS emissions because of measurement artifact unless you connect this explicitly to CPs for example. AUTHORS: This bit of historic information was originally connected to a paragraph about the gap in the current budget between sources and sinks, which is much bigger than the observed changes in atmospheric OCS concentration. Therefore, it is generally assumed that either sources are under- or sinks overestimated. However, that paragraph was discarded before submission. Without this context, this piece of information, while interesting, is probably also not required here and can be removed.

REFEREE: P3L3: Is there a paper arguing the missing sinks early on that you could cite here?

AUTHORS: As R1 has correctly pointed out, this statement is incorrect. Therefore, it will be removed. (Budgets currently are missing sources or the sinks are overestimated.)

REFEREE: P3L4: There is now a fair amount of literature on these drivers of OCS soil fluxes and mechanistic formulations representing them (see citations below). I'm sure this understanding can be tested in different soil, but is it fair to say the drivers are largely unknown? You cite additional papers to this point in the next few paragraphs, making that statement seem a bit contradictory. 1. Ogée, J.; Sauze, J.; Kesselmeier, J.; Genty, B.; Van Diest, H.; Launois, T.; Wingate, L. A new mechanistic framework to predict OCS fluxes from soils. Biogeosciences 2016, 13, 2221–2240, doi:10.5194/bg-13-2221-2016. 2. Sun, W.; Maseyk, K.; Lett, C.; Seibt, U. A soil diffusion–reaction model for surface COS flux: COSSM v1. Geosci. Model Dev. 2015, 8, 3055–3070, doi:10.5194/gmd-8-3055-2015. 3. Whelan, M. E.; Hilton, T. W.; Berry, J. A.; Berkelhammer, M.; Desai, A. R.; Campbell, J. E. Carbonyl sulfide exchange in soils for better estimates of ecosystem carbon uptake. Atmos. Chem. Phys. 2016, 16, 3711–3726,

doi:10.5194/acp-16-3711-2016. 4. Sun, W.; Maseyk, K.; Lett, C.; Seibt, U. Litter dominates surface fluxes of carbonyl sulfide in a Californian oak woodland. 2016, 438–450, doi:10.1002/2015JG003149.Received. 5. Kaisermann, A.; Ogée, J.; Sauze, J.; Wohl, S.; Jones, S. P.; Gutierrez, A.; Wingate,L. Disentangling the rates of carbonyl sulphide (COS) production and consumption and their dependency with soil properties across biomes and land use types. Atmos. Chem. Phys. Discuss. 2018, 1–27, doi:10.5194/acp-2017-1229.

AUTHORS: We agree that statement is not accurate. We propose to remove the sentence stating that drivers are largely unknown. Instead, after the list of known drivers with their citation we propose to add: "Unfortunately, the mechanistic pathways by which these drivers influence the soil-atmosphere exchange and the quantity of their impact are not always fully understood and quantified yet. Problems exist in view of the mix of contributing enzymes and microbial groups as well as the chemical quality of the soil, for example ammonium content and metabolism." As some of the soil samples we tested, contain high amounts of ammonium, we will specifically discuss the role of ammonium in our revised paper.

REFEREE: P3L11: Cite also Whelan et al., 2016

AUTHORS: Agreed, Whelan et al., 2016 should be cited here and will be added.

REFEREE: P3L20: These are net sinks, but there can still be gross production in forest soils (Kaisermann et al., 2018).

AUTHORS: While addressing a soil as a sink does not per se imply there is no production, it will be useful to include more details. We propose to change the sentence to (changes in bold): "Agricultural soils have been characterized as either a net OCS source or sink (Bunk, et al., 2017; Whelan et al, 2015; Maseyk et al., 2014), whereas forest soils have been characterized as net sinks (Sun et al., 2017; Steinbacher et al., 2004; Kaisermann et al., 2018).

REFEREE: P3L23: Would be helpful to distinguish that you mean 'atmospheric or headspace' $CO_2$ mixing ratios is what was controlled in that study (not pore-scale mixing ratios)

AUTHORS: Strictly speaking the referee is right and we will add this information. However, we would like to point out that the soil layer was very thin (several mm) in order to have all soil (unless covered by a water film) exposed to headspace conditions.

REFEREE: P5L15-27: The methods for calculating CP are circular and are not clearly explained. Please rewrite. Add additional equation notation to indicate the values determined where Eocs=0.

AUTHORS: Agreed. Additional equations will be added for the CP and kOCS as follows: kOCS= (EOCS,1000-EOCS,50)/(C1000-C50) (R1) where EOCS,1000 is the net OCS exchange at 1000 ppt OCS mixing ratio, EOCS5,0 is the net OCS exchange at 50 ppt OCS mixing ratio, C100 is the exact OCS mixing ratio when 1000 ppt OCS mixing ratio were set and C50 is the exact OCS mixing ratio when 1000ppt OCS mixing ratio were set. The CP is reached when EOCS is zero. This is the case when UOCS = POCS. Derived from eq(3) and (4) this can be expressed as: kOCS*CCP=POCS (R2) where CCP is the OCS concentration at which the compensation point at a given soil moisture. Solved for CCP we get: CCP = P_OCS/k_OCS (R3)

REFEREE: P5L28: I would clarify why the deposition velocity contains also the production term. This is for comparability to field measurements, correct?

AUTHORS: Yes, we are always measuring the net exchange which is comprised by both terms

REFEREE: P6L4: "Bourtsoukidis et al. (submitted)" is an unnecessary reference for this volumetric formulation. Please use a more suitable source. Define all terms in the equation (e.g., ti, ts)

AUTHORS: The definition of the terms will be added where needed. Bourtsoukidis
et al. is now published (Bourtsoukidis, E., Behrendt, T., Yañez-Serrano, A.M., Hellén, H., Diamantopoulos, E., Catão, E., Ashworth, K., Pozzer, A., Quesada, C.A., Martins, D. and Sá, M., 2018. Strong sesquiterpene emissions from Amazonian soils. Nature Communications). The deviation of the soil moisture equation we used can also be found in the supplement of Behrendt et al., 2014.

REFEREE: P6L18: If based on instrument precision, why would the resolution threshold vary by soil type? How is this different from your reported precisions L21, that have similar values?

AUTHORS: This noise is not purely dependent on the instrument precision. The experimental setup and scatter of the soil exchange also add to it. Therefore, it is a little higher than the precision in L21. The forest soils exhibited a stronger scatter, most likely because of their much higher activity (much higher uptake and emission rates).

REFEREE: P6L21: How did you determine the absolute mole fraction for the permeation source standard and for the certified gas mixtures? Which OCS source was used for which OCS mole fractions? What were their precisions? A supplemental figure showing the calibration curve should be shown to indicate how two of three types of calibration gases had excellent instrument response matching, but that the most high precision standard at atmospheric values was off by 7%.

AUTHORS: For the certified mixture the mixing ratio given by the manufacturer and the flow rate of the gas mixture and the synthetic air (both controlled by MFC) were used to calculate the expected mixing ratio. This commercial mixture has a higher uncertainty ($\pm$ 10 %) than the NOAA standard, which may have masked an underestimation of 7 % during our calibration measurements. For the permeation device, the permeation rate is calculated from the weight loss of the permeation container over a long observation time at closely monitored (and constant) air flow rate and temperature. The Forschungszentrum Jülich has established a setup for that purpose that is also cross checked against other methods. Marc von Hobe and his team kindly did a joint calibra-

tion check and inter-comparison of instruments (they have three LGR OCS Analyzers of their own) together with us, using their very sophisticated permeation setup.

REFEREE: P7L13: Is this gross uptake reported at the soil moisture where net uptake is highest? Clarify for your readers. 'Total uptake' is not clear. Please pick a set of terms for each meaning and stick to them throughout, such as: net exchange, net source, net sink, gross uptake, and gross production. Currently many terms are used in the text and figures (e.g., net release, exchange rate, total uptake, etc...).

AUTHORS: Yes, this is gross uptake. We agree this should be clarified and a set of terms defined in the beginning applied strictly throughout the text without using any other. This will be added/changed for the revision

REFEREE: P7L15: The confidence intervals or standard deviation on these average emissions (not ranges) can be given or statistical tests performed to show if they are different from zero.

AUTHORS: We will include a more detailed statistical discussion on that question in the revised manuscript.

REFEREE: P7L20-30: This paragraph could be condensed significantly to make the main point (intermediate OCS mixing ratios give intermediate fluxes, which indicate compensation points).

AUTHORS: We will try to compress the paragraph without removing too much of the information.

REFEREE: P9L24-31: The argument here about higher heterotroph abundance and activity being related to higher UOCS is not well supported by the data in this study or any references listed. The cited literature is not referenced appropriately (e.g., Kato 2007 was looking at mycobacteria, which are heterotrophs). This is a poorly developed line of reasoning, and I would suggest sticking to the chemical data in Table 2 to discuss possible drivers. I feel that this discussion point relies too strongly on submitted

manuscripts that can't be accessed by reviewers, inaccurately referenced studies, and statements without any references. Instead, the discussion should be de-scoped to accurately integrate published results alongside the data from this work.

AUTHORS: Based on reviewer input and reconsiderations we have decided to remove the discussion about heterotrophs and autotrophs as well as strongly reducing most discussion of microbial communities. Instead, other options as physical and chemical soil properties shall be included in the discussion of the revised version. Especially a topic of diffusion limitation by high soil moisture and a brief discussion of the possible role of ammonium oxidizing organisms, as there are significant differences in ammonium content in the examined soils that coincide with curve shapes of UOCS and kOCS, will be included. Also, the section can be shortened as it is covered in our follow up study which is already under review. We propose a merger of the current sections 4.2 and 4.3 (as kOCS and EOCS are necessarily tightly related) by compressing the points discussed therein and adding additional points of view (chemical and physical soil properties, diffusion limitation). Kato et al. (2007) is indeed incorrectly referenced and all references in this section will be carefully rechecked.

REFEREE: P10L14-22: The scope of this discussion point regarding production is much more in line with the paper (in contrast to aforementioned uptake sections) and seems quite appropriate.

AUTHORS: Thanks.

REFEREE: P11L8: The role of diffusion limitation of substrate at high soil moisture levels should be discussed. Enzymes are likely active, but substrate limited in that case. This statement is not supported: "We suspect these processes to be connected to autotrophic organisms, as discussed in Section 4.4."

AUTHORS: a discussion about diffusion limitation will be added (see above). Section 4.4 will be reduced in scope and probably be fused with another section. However, we think the possibility should be mentioned.

REFEREE: P11L10: The relationship between Eocs (and correspondingly Uocs shifted by the magnitude of Pocs) does not seem linear with OCS mole fraction, but instead increases significantly between 50 and 500 ppt and then less so from 500 to 1000 ppt. Please show a plot of the fits. What is a 'true trend'? If the relationship is not linear, errors in estimating U at E=0 may lead to errors in estimated kocs from two-point measurements. Are you confident this was not the case? This will bear on the discussion in 4.4

AUTHORS: We agree that there are some problems with the CP calculation that will be thoroughly discussed in the revised version. Our reply to R1s concerns applies here too: We agree that the method used to derive compensation points in this study may be improved by investing much more measuring time and sample material. However, we can rely on older reports about the linearity of OCS exchange between 50-1000 ppt in soils and plants. More data points to fit a regression would certainly be helpful. But it will further our understanding to present the current data set, also in view of the different soil types and chemical properties, and use it for discussion and development of new approaches. The limitations will definitely be addressed in a revised manuscript. However, our large data set provides a consistency over the whole range of the investigated soil water content, which covers a much wider range of soil moisture than in previous reports. To address the uncertainties, we will add error estimations based on Gaussian error propagation for the net release (EOCS), uptake coefficient (kOCS), gross uptake (UOCS), gross production (POCS) and compensation points (CP). Of further importance is the question about linearity of the relationship between OCS concentration and exchange rates at higher OCS mixing ratios. Earlier reports demonstrate good linearity up to approximately 1500 ppt OCS ambient concentration in measurements with soil (Kesselmeier et al., 1999). Investigations performed with lichens (Kuhn and Kesselmeier, 2000), which are lacking any outer protective cell layers and seem to behave similarly as soils, exhibit a comparable linearity relating exchange rates and OCS concentrations. In both cases the authors report a saturation effect at higher OCS concentrations, which leads to an increasing uncertainty. The reasons are not
investigated but can be understood as metabolic saturation (enzyme saturation) or different/additional biotic processes influencing the rate of OCS exchange. These uncertainties limit the reliability of the compensation points calculated, especially if those are >1500ppt or at soil moistures above 60 % WFPSlab. However, despite this uncertainty, the CPs show a clear and consistent trend that cannot be disregarded. It can clearly be seen that the uptake coefficient (slope of the regression) first increases and then decreases again with decreasing soil moisture. Setting the reliability of the absolute values aside, such coherent trends contain information that can be explored. Additionally, the calculated CPs have reasonable overlap with older data. An important aspect of the paper is the observation of CPs over soil moisture which has not been reported before. Even with relatively high uncertainties this data is worth to pursue and to plan future experiments accordingly. For the current revision we propose to test the significance of all trends in EOCS, kOCS and POCS by using the first data point of each measurement as control and perform a t-test to see if the next value is statistically different from the previous one. It would highlight the idea of monitoring a significant trend in OCS over drying out.

REFEREE: P12L8: Is there a specific heterotrophic CO2 fixation reference you can provide for the relation to OCS? The paper should focus on fluxes and CPs you observed without too much speculation on microbial metabolisms implicated. As stated by the authors, that activity was beyond the scope of the work.

AUTHORS: As said above, we agree that more room should be given to discussion of the physical and chemical factors that may influence the fluxes and CPs and will add this in the revised version. We have decided to abandon the discussion about autotrophy/heterotrophy.

Referenced Literature:

Ogée, J., Sauze, J., Kesselmeier, J., Genty, B., Van Diest, H., Launois, T. and Wingate, L., 2016. A new mechanistic framework to predict OCS fluxes from soils. Biogeosciences, 13(8), pp.2221-2240.

Whelan, M.E., Lennartz, S.T., Gimeno, T.E., Wehr, R., Wohlfahrt, G., Wang, Y., Kooijmans, L.M., Hilton, T.W., Belviso, S., Peylin, P. and Commane, R., 2018. Reviews and syntheses: Carbonyl sulfide as a multi-scale tracer for carbon and water cycles. Biogeosciences, 15(12), pp.3625-3657.

Kaisermann, A., Ogée, J., Sauze, J., Wohl, S., Jones, S. P., Gutierrez, A., and Wingate, L.: Disentangling the rates of carbonyl sulphide (COS) production and consumption and their dependency with soil properties across biomes and land use types, Atmos. Chem. Phys. Discuss., https://doi.org/10.5194/acp-2017-1229, in review, 2018

Kuhn, U. and Kesselmeier, J. (2000) Environmental variables controlling the uptake of carbonyl sulfide by lichens. J. Geophys. Res. 105 (22), 26783-26792. Bunk, R., Behrendt, T., Yi, Z., Andreae, M.O. and Kesselmeier, J., 2017. Exchange of carbonyl sulfide (OCS) between soils and atmosphere under various CO2 concentrations. Journal of Geophysical Research: Biogeosciences.

Bourtsoukidis, E., Behrendt, T., Yañez-Serrano, A.M., Hellén, H., Diamantopoulos, E., Catão, E., Ashworth, K., Pozzer, A., Quesada, C.A., Martins, D. and Sá, M., 2018. Strong sesquiterpene emissions from Amazonian soils. Nature Communications

Behrendt, T., Veres, P.R., Ashuri, F., Song, G., Flanz, M., Mamtimin, B., Bruse, M., Williams, J. and Meixner, F.X., 2014. Characterisation of NO production and consumption: new insights by an improved laboratory dynamic chamber technique. Biogeosciences, p.5463.

---

## Author Comment (AC3) · 28 Jun 2018

REFEREE: This is an interesting study that investigates how the net carbonyl sulfide (COS) exchange between a set of soils (forest and agricultural) and the atmosphere is composed of two opposing component fluxes (a gross uptake flux of COS and a gross production flux of COS) that are regulated differently in response to variations in soil water filled pore space. They revisit the compensation point method of Conrad, 1994 and Lehmann & Conrad, 1996 to obtain estimates of COS production from observations of the net soil COS exchange measured under two different atmospheric COS concentrations of 50 and 1000 ppt. These concentrations are much lower than

those used in Lehmann & Conrad, 1996 and likely more applicable to concentrations observed in the field. Although many studies have measured and modeled from theory the response of the net COS exchange to variations in soil moisture for a given soil and how differences in soil texture and bulk density play on the WFPS and subsequently the optimum net COS exchange (i.e. the maximum COS uptake rate measured), this is the first study to show that the COS production rate remains more or less constant over the entire range of %WFPS. Thus variations in the net COS exchange with WFPS are driven by the uptake component of the net COS flux. Between the different soils measured (3 spruce and 1 agricultural site) there were large differences in the magnitudes of the net COS flux and their component fluxes with largest fluxes for all components found in the Finnish forest and the lowest component fluxes observed in the agricultural soil in Germany. In addition a further experiment showed that after drying the agricultural soil for several months and re-humidifying, a very similar response of the net and component fluxes to % WFPS was observed, although no statistics were completed to test whether they were significantly different or not. In general there is some nice data in this study that is definitely worthy of publication.

AUTHORS: We thank the referee for his review, the good suggestions made and invested work.

REFEREE: Major comments Throughout the manuscript there is very ambiguous application of terminology regarding the exact flux being presented. When the paper is expressly about partitioning the components of a net flux, one has to take care to be precise and state clearly which flux they are writing about. I thoroughly recommend that the authors go through the paper and clarify exactly what each flux is that they refer to, when they refer to it in the paper. Simply referring to COS exchange is too ambiguous, this paper must always refer to the net COS flux (EOCS), the COS emission rate or production rate (POCS) and the COS uptake rate (UOCS).

AUTHORS: We agree clear terminology is very important here and will revise accordingly. All instances will be either be net release (EOCS), gross uptake (UOCS) or gross

production (POCS)

REFEREE: Furthermore, the partitioning approach taken in the current study does not completely isolate the two component gross fluxes, rather the uptake term measured at a constant temperature as presented by the authors is still regulated to some extent by diffusion (not strictly enzymatic uptake of COS) and the production term as presented still incorporates a COS deposition velocity (Vd0) that occurs even when the COS concentration = 0. These details are developed in the Ogee et al., 2016 paper https://www.biogeosciences.net/13/2221/2016/bg-13-2221-2016.pdf and more relevantly to the current study in a recent publication in Atmospheric Chemistry and Physics Discussions by Kaisermann et al. https://www.atmos-chem-physdiscuss.net/acp-2017-1229/acp-2017-1229.pdf that demonstrates within the current methodological framework that a small additional analysis is required to obtain the gross COS production rate when COS concentration = 0 that must be solved iteratively see Eqs 2, 3 and 4 of Kaisermann et al.

AUTHORS: We agree that the most accurate POCS values would be obtained with the method described in Kaisermann et al., 2018. However, when we estimate the changes this correction would make to POCS in this study, these changes are below the uncertainty of obtained values: Derived from eq2 and eq3, we can say that the gross uptake (UOCS) is the consumption rate coefficient (kOCS) times the ambient OCS concentration (C). →UOCS= kOCS x C. For a rough estimation how much POCS will maximally be changed by Vd0 we can assume that the concentration C0 that the sinks in the soil are exposed to when the mixing ratio in the flushing air is zero can be derived from the (uncorrected) production rate. With the median production rates of 2, 4, 7 pmol g-1 h-1 and maximum kOCS of -0.005, -0.0015, -0.0013 and -0.0146 mol g-1 h-1 for Mainz soil, Waldstein soil with spruce or blueberry understory, and Finland needle forest litter, respectively, we get an estimated correction of 0.010, 0.011, 0.005 and 0.394 pmol g-1 h-1 These corrections would be smaller than the uncertainty of our measured exchange rates. As we do not have the accurate soil depth (zmax) and

would have to use estimates (based on filling measurement chambers again with the amount of sample used in the experiments and measure soil depth), we think we can neglect this correction. Of course, the topic of these corrections will still be discussed in the revised version.

REFEREE: Another important point that was expressed several times in the paper is that the shape of the net COS exchange response to WFPS is unknown but probably caused by changes in the activity of the enzyme CA. However, this is not strictly true as in the past few years the community has made considerable progress in explaining the response of the net COS flux to variations in soil temperature, soil moisture, soil texture and soil microbial biomass collectively in the papers of Kesselmeier et al., 1999; Van Diest &Kesselmeier, 2007; Ogee et al., 2016 and Sun et al., 2016. From these papers it has been shown that the observed optimum with soil moisture content observed in the present study and many times in the literature can be modelled extremely well and is mostly caused by changes in the diffusion of OCS within the soil matrix that reduces the potential hydrolysis rate at a given temperature, microbial biomass and COS concentration. This can occur over the typically short time frames of these experiments and thus the net COS flux and the gross COS uptake rate does not need to be driven by changes in the intrinsic enzyme activity or size of the microbial population. Thus the discussion needs to take this in to account and furthermore considered in the interpretation of the data presented in the results. Subsequently, differences in soil texture can probably explain most of the differences in the absolute %WFPS values where the optimum net COS flux is observed. Unfortunately, no data is provided in the manuscript about the differences in texture between the sites, this should be added. As described above the data in this study are interesting however the presentation of the results could definitely be improved and synthesised.

AUTHORS: We agree that these aspects are underrepresented in the current version of the manuscript. We will cut back the deliberations about microbial communities and enzyme activity in the revised version and will scrap the discussion about heterotrophs/autotrophs altogether. Instead we will expand the discussion about diffusion limitation by soil moisture, chemical properties of the soil (especially ammonium content). Unfortunately, only limited texture information is available.

REFEREE: I see no reason why figures 1 and 2 are not merged and a more synthetic analysis of the fresh vs dried/re-humidified soil results presented in a new Figure 2. This new figure could consist of 3 panels side by side. In panel (a) the authors would present the data from the 50ppt experiments, panel (b) the 500 ppt and panel (c) the 1000 ppt data. On the x-axis of each panel would be the net COS flux from the fresh soil against axis y the net COS flux from the same re-humidified dry soils. The authors could then colour the points by %WFPS (light blue = dry soils and dark blue = wet soils). Then they could also show the 1:1 line and do some regression statistics that way the audience can assess clearly and objectively the effect of the drying on the %WFPS response. The authors also point out that the consumption rate coefficient (k) follows the same pattern with %WFPS as the net COS flux and the partitioned COS uptake rate (Uocs) and present the variability of k with %WFPS. However, U and k must vary with soil moisture and temperature... etc as they are linearly related or even proportional if Vocs is always at the same COS concentration. Thus I would remove figure 5 and rewrite the discussion to address this point.

AUTHORS: We agree that Figure 1 and 2 can easily be merged and will do that. We thank the referee for the very interesting suggestion for a new comparison figure dry vs. fresh and will adapt this.

REFEREE: Also the compensation point does not affect the COS exchange rate and thus the title of the manuscript should be corrected.

AUTHORS: Yes, we agree that the title is misleading and will change it.

REFEREE: The authors state that the data from the Sun et al 2017 field study and this study are comparable and can be used to transfer the findings from the lab data to the field. However, the lab response to soil moisture content (green line) does not

go through the middle of the points, but rather forms an upper envelope and there is no statistical test behind this statement. At the minimum the authors should calculate the mean deposition velocity for the relevant and comparable soil moisture values for their study and the Sun et al study and compare the means. Furthermore, I do not think it is appropriate to use the temperature optimum from the Mainz soil to make the field fluxes of the Finnish soil comparable. It has been demonstrated before that the net COS flux is strongly affected by the production rate and can cause a shift in the temperature optimum. As this study shows the Mainz and Finnish soils have extremely different production rates I do not think this is the most appropriate way to reconcile the two data sets and facilitate comparison.

AUTHORS: We agree that averages or a fit are a good idea and will implement this. We think that using the temperature curve from Kesselmeier et al., is the best available approximation. Another approach would be to follow the Kaisermann et al., 2018. They found 1.23 for kOCS Q10 value for 10° C and a large set of different soils. So for 5° C it should be 0.615. Either method is only an approximation, but in our opinion clearly better than no temperature correction at al. Naturally a discussion of the weaknesses of the correction method will be added.

REFEREE: Specific comments Page 3 of the introduction lacks a number of citations that describe the theoretical advances the research community has made in describing the response of net COS exchange to soil water, texture, soil temperature etc... there is also some internal contradiction within the text. Finally the introduction does not present any hypotheses on why they might expect shifts in sources and sinks relative to their experimental manipulation with COS concentration and soil moisture content.

AUTHORS: We will add a section about the theoretical advances, especially Ogee et al., 2015. A paragraph of the relationship of OCS uptake and ambient OCS concentration will be moved from its current position into the introduction, as has been suggested by another reviewer. We will also add research questions to the introduction, as detailed in our reply to R1.

REFEREE: The characteristics of the soils provided are partially useful. It would be better to provide the physical characteristics such as bulk density or texture rather than nitrate and ammonium fluxes taken from some other studies and not conducted on the soils at the same time as when measured. These values could be misleading as inorganic N concentrations are turned over rapidly and vary with season and management.

AUTHORS: We assume the referee means nitrate and ammonium contents as no fluxes are presented here. We would like to point out that our samples and the samples for the cited studies came from the same sample pool (out of the same bag of soil sample) including the measurement of N contents. Samples were stored at 4°C in the dark and sub-samples drawn for various studies. Therefore, we do not think the referees' concerns apply here. The bulk density will be added to the table.

REFEREE: Pg 4 line 10 what exactly can the author not exclude variability in over time? Was there no fixed protocol for the collection, storage and handling of the soils?

AUTHORS: A common protocol was in place. But the sampling was done by different persons and some of the steps contain a degree of subjectivity. For example, how long and vigorous a sieve is shaken influence how much of the particles sized close to the mesh size pass through. Similarly, the practical decision on the top 5 centimeters of soil contains some subjectivity. However, the problem is probably overstated. We think the statement about variability over time can be removed.

REFEREE: Pg 4 line 14 length of sample storage should be provided here and was it the same for each soil?

AUTHORS: we will add when each sample was collected (Mainz soil: January 2014, the other soils fall 2012). All experiments were performed in February 2014.

REFEREE: Section 2.4 and description of %WFPS protocol should come just after Section 2.1

AUTHORS: this change in sequence is a good idea. We will do that.

**BGD**

REFEREE: Pg4 line 20 state the temperature of the soil here and how constant.

AUTHORS: Soil temperature was 20° C. Temperature variation was less than 0.5° C. The information will be added.

REFEREE: Pg 4 Section 2.2 how many sample replicates are measured for each soil at a time and what do the error bars refer to in the figures?

AUTHORS: The error bars denote the standard deviation of 30 second averages. There have been no replicates. However, some experiments were done multiple times under similar conditions and we found that the resulting exchange over soil moisture was always very similar, though there were several months between these measurements and the soil chamber was moved to another lab and build up again. See the attached figure originally from Bunk et al, 2017.

REFEREE: Section 2.2 How long between wetting and gas exchange started? How long is the measurement sequence? How long is the airstream sampled? How do you check for steady-state? Information on when the soils were sampled would be useful e.g. time of year; before/after fertilization?

AUTHORS: The incubation and sampling technique was described in Bunk et al. (2017). We repeat it here for a better understanding. One measurement sequence (from 100 to 0 % soil moisture) is about 60 hours. Flushing of the chambers started about 1-2 minutes after wetting (the time it takes to screw the chamber tight). Chambers were measured in sequence for 10 Minutes. This means all chambers were constantly flushed and their outlet connected for 10 minutes to the analyzer each 40 minutes (3 samples plus one empty reference chamber). Our setup does not use true steady state conditions (which is not possible with a dynamic chamber approach) but a dynamic equilibrium that is for all purposes of this study analog to true steady state conditions: In a well-mixed chamber (ensured by the activity of a fan in the headspace of each cuvette in our setup) the concentration of a trace gas is determined by the concentration in the flushing gas, the rate of flushing and the release or uptake by the

soil. The flushing rate is constant and monitored. Factors like soil moisture are very slow changing in relation to the measurement period averaged for calculation by eq1. Chamber effects are accounted for by using an empty identical chamber for reference. While true steady state conditions can only be archived with static chambers, a dynamic equilibrium was archived in flushed chambers that fulfills all conditions required to employ eq1. This dynamic equilibrium was ensured as described below and can/will be involved in the revised paper: Switching the analyzer inlet from one chamber to the next of course courses some disturbances in the chamber and also the inlet tube needed flushing. We did extensive pre-measurements by observing measured chamber concentration with chambers that were empty or loaded with soil samples, waiting for steady concentrations to be measured. With the length of inlet tube and flushing rate used, stable concentrations were always observed after less than six minutes. Based on these empirical determinations, the time each chamber was connected to the analyzer was set to ten minutes. Of these ten minutes, the first 7 minutes were discarded to allow for proper equilibration within the chamber and at the same time allowing proper flushing time for the analyzer and analyzer inlet tube. The following 2.5 minutes were averaged for 30 seconds intervals and used for exchange calculations. The last 30 seconds of each chamber cycle was discarded again to eliminate any chance of accidental overlap. Information about the sampling year and other available information about the soil sampling time and conditions will be added.

REFEREE: Section 2.4 remove the citation Bourtsoukidis et al

AUTHORS: Bourtsoukidis et al. were the ones from whom the method was adapted. The paper is now published so we think they should be cited.

REFEREE: Page 6 Line 24 is it the fluxes or absolute mixing ratios underestimated by 7%?

AUTHORS: The absolute mixing ratios. But since the fluxes are derived from the difference between the sample and reference mixing ratio, that propagates to the fluxes

as well.

REFEREE: Page 7 line 12 I feel the last sentence of this paragraph is out of place and should appear later.

AUTHORS: We respectfully disagree. EOCS and POCS are described here, why would UOCS be out of place?

REFEREE: Section 3.2 really needs to be re-written. It is ambiguous, difficult to read and contains repetition. In places it is hard to work out what the authors are comparing.

AUTHORS: We will find a clearer phrasing.

REFEREE: Page 8 line 3 you should state that the soils had compensation points higher than the background atmosphere.

AUTHORS: We agree and will do that.

REFEREE: Section 3.3 again some repetition at the end of the paragraph. Also the authors should point out that compensation points in themselves are not particularly useful as they are not intrinsic properties of a soil as their value will vary with temperature and COS concentration. They are only useful for establishing whether a soil is a source or sink of COS to the atmosphere.

AUTHORS: We partially agree. The last bit of the paragraph can be pruned as it is said before. It is also true that CPs can only give limited information because they are co-dependent from many factors. However, such a discussion belongs in the discussion section. We will add it there.

REFEREE: Section 3.4 the figure 4 panel b production rate looks very strange. I am not sure why it has this appearance, but I am guessing maybe there is some interpolation being made between a limited number of measurements over the drying curve. It would be useful to explain what is going on here in the methods, results and the legend

AUTHORS: We agree that the production rate for this sample looks a bit strange. However, we could not find any indication that something is out of place. At around 50% WFPS the scatter of data points for the 50ppt net release (EOCS) measurement is a bit increased. This coincides with the "out of place" peak for the calculated production.

REFEREE: Page 8 line 19-20 This statement as described above is redundant as is the figure 5 and is not linked to any of the things proposed in Section 4.2

AUTHORS: As stated above we agree this sentence can be pruned.

REFEREE: Page 8 line 30 this correction may have some problems associated with it, if the authors insist on using it they should be more critical about why it is not ideal.

AUTHORS: We agree that this correction is not ideal. A discussion of the potential problems is warranted and will be added. However, we think it is the best possible approximation (see also comments to referee 1).

REFEREE: Page 9 Ln 3 the reason for the scatter is because many variables are changing at the same time, please be a bit more critical.

AUTHORS: Yes, this was also our conclusion. That is why we wrote in line 5: "We suggest that the stronger scatter in the field measurements is due to additional factors that were kept constant in our lab measurements, but unavoidably vary during field measurements". Of course we will elaborate a bit more on that point.

REFEREE: Page 9 Ln 8 I would not hold this graph up as evidence that your study can simulate what is happening in the field.

AUTHORS: We disagree. Observed deposition velocity not only is very close in absolute value but also shows the same trend as the averages of the reported field data. This should become even clearer after modifying figure 6 as suggested by R2.

REFEREE: Page 9 line 11-22 This discussion is a little imaginative and is not so relevant to the results. The arguments do not follow a clear logic.

AUTHORS: We have decided to abandon this line of reasoning, instead focusing mainly

on diffusion limitation, effects of the high ammonium content in the Waldstein soil and other physical and chemical properties.

REFEREE: Page 10 Line 11-13 I don't understand these statements. What exactly has it's activity reduced? What is the different uptake process and what evidence in your data do you have for this?

AUTHORS: That was referring to the aforementioned consumers. However, we propose to only point out there is a follow up study where microbial methods were employed. As we also decided to give up the discussion about heterotrophs and autotrophs in this study, we propose to prune away the lines 7 to 11.

REFEREE: Page 10 line 22 What about abiotic processes?

AUTHORS: "All known and unknown OCS production pathways" was supposed to include abiotic processes. We will revise to "All known and unknown OCS production pathways and processes".

REFEREE: Page 10 line 28-30 ambiguous statements about exchanges and observed values without being precise about what they are referring to. Which exchanges and values exactly?

AUTHORS: For the Waldstein Blueberry soil, the net exchange at 50 ppt is so close to 0 that, statistically, many data points cannot safely be distinguished from a zero exchange (the standard deviation of the sample and the reference chamber mixing ratios overlap). However, for all moistures, there are still some points for which the difference between the reference and sample mixing ratio exceeds the uncertainty. The data points where the difference between the sample- and reference mixing ratio is not bigger than the uncertainty still follow the same trend as the data points where the difference in mixing ratio does exceed uncertainty. Additionally, the Waldstein Blueberry soil shows the same general trend as the other soils at low ambient mixing ratios. We will revise to clarify the reasoning.

REFEREE: Page 11 section 4.2 lines 7-15 I don't think any of these arguments are relevant to the results they are discussing. The authors do not appreciate the role of diffusion within the soil matrix and it's affect on the ability of soils to take up COS or not as WFPS changes. This should be explored first before jumping to the conclusion that autotrophic organisms have some role to play in explaining this pattern, especially as the authors have no evidence to support this hypothesis.

AUTHORS: Based on reviewer input and reconsiderations we have decided to remove the discussion about heterotrophs and autotrophs as well as strongly reducing most discussion of microbial communities. Instead, other options as physical and chemical soil properties shall be included in the discussion of the revised version. Especially a topic of diffusion limitation by high soil moisture and a brief discussion of the possible role of Ammonium oxidizing organisms, as there are significant differences in Ammonium content in the examined soils that coincide with curve shapes of UOCS and kOCS, will be included.

REFEREE: Section 4.2 last paragraph should be in the results section it is not discussion.

AUTHORS: We will move the paragraph to the results section

REFEREE: Page 12 line 3 also mention the other factors that will alter the compensation point.

AUTHORS: Agreed.

REFEREE: Page 12 line 14-16 your experimental data does not support this statement about the two compensation points of Lehmann & Conrad please modify the sentence.

AUTHORS: We will remove this sentence.

REFEREE: Page 13 Ln 2-19 this discussion again contains a lot of conjecture and fails to mention that the differences in texture between the spruce sites is probably important and should be accounted for before attributing differences in COS uptake rates to other

factors

AUTHORS: We have decided to completely abandon the heterotroph/autotroph reasoning as not supportable with our data. Instead it will be discussed in the view of diffusivity and soil properties, especially Ammonium content (which is very high for the Waldstein soils). Also a topic of consideration will the origin layer (topsoil vs organic layer).

REFEREE: Page 13 line 23 I think this is important info that should appear in the methods section.

AUTHORS: The sampling year will be added for all soils to their description in the methods section and Table 1.

REFEREE: Page 13 line 28-30 don't you think this is because fundamentally the soil texture did not change over the 6 years at the Mainz site and thus you observe a similar pattern when you wet and dry the soil 6 yrs later?

AUTHORS: That's the point. The soil was sampled, then sampled again six years later. We get very similar patterns, so neither the physical properties nor the sink strength (determined by the microbial communities inside the soil) can have changed too much. And that is something that is good to know, as many budget calculations rely on soil studies that have only been done once. It may mean that agricultural soils do not change that much, unless their crop-cycle or fertilization mode is changed.

REFEREE: Page 13 line 18 LAI of 15? I don't believe this is possible

AUTHORS: This is the value given by the cited authors. Any debate of the accuracy of this value would have to be taken up with them. Besides, even if that number is not accurate it is still undisputed that the LAI can vary a lot from site to site. Either way, section 4.6 will be removed due to its tangential nature.

REFEREE: Section 4.6 I think this is a bit long and not so useful in the end I think it could be summarized in a sentence or two in the conclusion Conclusions are currently

based on conjecture and not the results. Statement 1 The experiment was not designed to test and cannot prove that COS is driven by the litter layer. Statement 2 There are no data presented about fungi in this paper so again this statement seems redundant. Statement 3 They do not prove that COS uptake is driven by different enzymatic processes. Statement 4 Is an introduced error of 1% really significant? Statement 5 No evidence in this study that the correlation coefficient k is linked to the presence or absence of auto- or heterotrophic organisms. Statement 6 I agree trying to understand compensation point variability without a model that accounts for how it varies with T, moisture and COS concentration is frustrating. We should use the theory and models that now exist to address this issue. Statement 7 they did not demonstrate statistically that the storage issues introduced significant differences in the fluxes and what level of uncertainty is introduced.

AUTHORS: Based on the input by R1, R2 and R3 we decided to remove section 4.6. The main message will be added condensed to a few sentences at an appropriate point. The conclusions might be a good place for that, as the referee suggests. Based on referee feedback, the discussion will change quite a bit and the conclusions will be revised accordingly.

REFEREE: Merge Fig 1 and 2

AUTHORS: we will do that

REFEREE: Recommend a synthetic figure 2 with some statistical analysis.

AUTHORS: We will add a figure as suggested in the reviewer's general remarks. Thank you again for the suggestion.

REFEREE: Figure 3 should there not be some estimation of error on these points?

AUTHORS: The error of these points will be discussed within the text of the revised version and a reference to that discussion added to the figure caption in the revised version. Error bars will be added if they do not clutter the figure too much.

REFEREE: Figure 4 panel b looks weird also can you show that the inlet is constant during each of the experiments and that steady state was attained

AUTHORS: Please see our comment on the topic of steady state above. Regarding the inlet, inlet flow rate was logged and is constant.

REFEREE: Figure 5 redundant Figure 6 not sure this is necessary either

AUTHORS: We disagree

REFEREE: Figure 7 not sure this figure is explicitly referred to in the text or necessary in the paper.

AUTHORS: The figure is referred to in section 4.5

REFEREE: Table should state explicitly which nitrate and ammonium data are relevant to the gas exchange measurements taken during the actual present study experiment.

AUTHORS: please refer to our reply to your major comment section above.

Referenced literature:

Kaisermann, A., Ogée, J., Sauze, J., Wohl, S., Jones, S. P., Gutierrez, A., and Wingate, L.: Disentangling the rates of carbonyl sulphide (COS) production and consumption and their dependency with soil properties across biomes and land use types, Atmos. Chem. Phys. Discuss., https://doi.org/10.5194/acp-2017-1229, in review, 2018

Bourtsoukidis, E., Behrendt, T., Yañez-Serrano, A.M., Hellén, H., Diamantopoulos, E., Catão, E., Ashworth, K., Pozzer, A., Quesada, C.A., Martins, D. and Sá, M., 2018. Strong sesquiterpene emissions from Amazonian soils. Nature Communications

Bunk, R., Behrendt, T., Yi, Z., Andreae, M.O. and Kesselmeier, J., 2017. Exchange of carbonyl sulfide (OCS) between soils and atmosphere under various CO2 concentrations. Journal of Geophysical Research: Biogeosciences.

Ogée, J., Sauze, J., Kesselmeier, J., Genty, B., Van Diest, H., Launois, T. and Wingate,

L., 2016. A new mechanistic framework to predict OCS fluxes from soils. Biogeosciences, 13(8), pp.2221-2240.

[Figure]

[Figure]

**Fig. 1.** Three measurements of EOCS at 1000 ppt OCS and 440 ppm $CO_2$ under similar conditions in three different experiments. Fingure originally from Bunk et al., 2017